# CorrAttack: Black-box Adversarial Attack with Structured Search

## Abstract

We present a new method for score-based adversarial attack, where the attacker queries the loss-oracle of the target model. Our method employs a parameterized search space with a structure that captures the relationship of the gradient of the loss function. We show that searching over the structured space can be approximated by a time-varying contextual bandits problem, where the attacker takes feature of the associated arm to make modifications of the input, and receives an immediate reward as the reduction of the loss function. The time-varying contextual bandits problem can then be solved by a Bayesian optimization procedure, which can take advantage of the features of the structured action space. The experiments on ImageNet and the Google Cloud Vision API demonstrate that the proposed method achieves the state of the art success rates and query efficiencies for both undefended and defended models.

## 1 Introduction

Although deep learning has many applications, it is known that neural networks are vulnerable to adversarial examples, which are small perturbations of inputs that can fool neural networks into making wrong predictions (Szegedy et al., 2014). While adversarial noise can easily be found when the neural models are known (referred to as *white-box attack*) (Kurakin et al., 2016). However, in real world scenarios models are often unknown, this situation is referred to as *black-box attack*.

Some methods (Liu et al., 2016; Papernot et al., 2016) use the transfer-based attack, which generates adversarial examples on a substitute model and transfer the adversarial noise to the target model. However, the transferability is limited and its effectiveness relies highly on the similarity between the networks (Huang & Zhang, 2020). If two networks are very different, transfer-based methods will have low success rates.

In practice, most computer vision API such as the Google Cloud Vision API allow users to access the scores or probabilities of the classification results. Therefore, the attacker may query the black-box model and perform zeroth order optimization to find an adversarial example without the knowledge of the target model. Due to the availability of scores, this scenario is called *score-based* attack.

There have been a line of studies on black-box attack which directly estimate the gradient direction of the underlying model, and apply (stochastic) gradient descent to the input image (Ilyas et al., 2018; 2019; Chen et al., 2017; Huang & Zhang, 2020; Tu et al., 2018; Li et al., 2019). In this paper, we take another approach and formulate score-based attack as a time-varying contextual bandits problem. At each state, the attacker may change the adversarial perturbation and get the reward as the reduction of the loss. And the attacker would receive some features about the arms before making the decision. By limiting the action space to image blocks, the associated bandits problem exhibits local correlation structures and the slow varying property suitable for learning. Therefore, we may use the location and other features of the blocks to estimate the reward for the future selection of the actions.

Using the above insights, we propose a new method called CorrAttack, which utilizes the local correlation structure and the slow varying property of the underlying bandits problem. CorrAttack uses Bayesian optimization with Gaussian process regression (Rasmussen, 2003) to model the correlation and select optimal actions. A forgetting strategy is added to the algorithm so that the Gaussian process regression can handle the time-varying changes. CorrAttack can effectively find

blocks with the largest rewards. The resulting method achieves much lower numbers of average queries and higher success rates than prior methods with a similar action space (Moon et al., 2019).

It is worth noting that BayesOpt (Ru et al., 2020) and Bayes-Attack (Shukla et al., 2019) also employ Bayesian optimization for score-based attack. However, their Gaussian process regression directly models the loss as a function of the image, whose dimension can be more than one thousand. Therefore, their speed is slow especially for BayesOpt, which uses slow additive kernel. CorrAttack, on the other hand, searches over a much limited action space and models the reward as a function of the low dimensional feature. Therefore, the optimization of CorrAttack is more efficient, and the method is significantly faster than BayesOpt.

We summarize the contributions of this work as follows:

1. We formulate the score-based adversarial attack as a time-varying contextual bandits, and show that the reward function has slow varying properties. In our new formulation, the attacker could take advantage of the features to model the reward of the arms with learning techniques. Compared to the traditional approach, the use of learning in the proposed framework greatly improves the efficiency of searching over optimal actions.

2. We propose a new method, CorrAttack, which uses Bayesian optimization with Gaussian process regression to learn the reward of each action, by using the feature of the arms.

3. The experiments show that CorrAttack achieves the state of the art performance on ImageNet and Google Cloud Vision API for both defended and undefended models.

## 2 RELATED WORK

There have been a line of works focusing on black-box adversarial attack. Here, we give a brief review of various existing methods.

**Transfer-Based Attack** Transfer-based attack assumes the transferability of adversarial examples across different neural networks. It starts with a substitute model that is in the same domain as the target model. The adversaries can be easily generated on the white-box substitute model, and be transferred to attack the target model (Papernot et al., 2016). The approach, however, depends highly on the similarity of the networks. If two networks are distinct, the success rate of transferred attack would rapidly decrease (Huang & Zhang, 2020). Besides, we may not access the data for training the substitute model in practice.

**Score-based Attack** Many approaches estimate the gradient with the output scores of the target network. However, the high dimensionality of input images makes naive coordinate-wise search impossible as it requires millions of queries. ZOO (Chen et al., 2017) is an early work of gradient estimation, which estimates the gradient of an image block and perform block-wise gradient descent. NES (Wierstra et al., 2008) and CMA-ES (Hansen, 2016) are two evolution strategies that can perform query efficient score-based attack Ilyas et al. (2018); Meunier et al. (2019). Instead of the gradient itself, SignHunter (Al-Dujaili & O'Reilly, 2020a) just estimates the sign of gradient to reduce the complexity. AutoZOOM (Tu et al., 2018) uses bilinear transformation or autoencoder to reduce the sampling space and accelerate the optimization process. In the same spirit, data prior can be used to improve query efficiency (Ilyas et al., 2019). Besides, MetaAttack (Du et al., 2020) takes a meta learning approach to learn gradient patterns from prior information, which reduces queries for attacking targeted model.

Many zeroth order optimization methods for black-box attacks rely on gradient estimation. However, there are some research works using gradient free methods to perform black-box attack. BayesOpt and Bayes-Attack (Ru et al., 2020; Shukla et al., 2019) employ Bayesian optimization to find the adversarial examples. They use Gaussian process regression on the embedding and apply bilinear transformation to resize the embedding to the size of image. Although the bilinear transformation could alleviate the high dimensionality of images, the dimension of their embeddings are still in the thousands, which makes Bayesian optimization very ineffective and computationally expensive. A different method, PARSI, poses the attack on $\ell_\infty$ norm as a discrete optimization problem over $\{-\varepsilon, \varepsilon\}^d$ (Moon et al., 2019). It uses a Lazy-Greedy algorithm to search over the space $\{-\varepsilon, \varepsilon\}^d$ to find an adversarial example. SimBA (Guo et al., 2018) also employs a discrete search space targeted at $\ell_2$ norm.

**Decision-based Attack** Decision-based attack assumes the attacker could only get the output label of the model. Boundary Attack and its variants (Brendel et al., 2017; Chen et al., 2020; Li et al., 2020) are designed for the setting. However, the information received by the attacker is much smaller than score-based attack, and it would take many more queries than score-based attack to successfully attack an image.

## 3 PRELIMINARIES

A Gaussian process (Rasmussen, 2003) is a prior distribution defined on some bounded set $\mathcal{Z}$, and is determined by a mean function $\mu : \mathcal{Z} \to \mathbb{R}$ and a covariance kernel $\kappa : \mathcal{Z} \times \mathcal{Z} \to \mathbb{R}$. Given $n$ observations $\mathcal{D}_n = \{(z_i, f(z_i))\}_{i=1}^n$, the prior distribution on $f(z_{1:n})$ is

$$f(z_{1:n}) \sim \text{Normal}(\mu_0(z_{1:n}), \kappa_0(z_{1:n}, z_{1:n})), \tag{1}$$

where we use compact notation for functions applied to collections of input points: $z_{1:n}$ indicates the sequence $z_1, \cdots, z_n$, $f(z_{1:n}) = [f(z_1), \cdots, f(z_n)]$, $\mu_0(z_{1:n}) = [\mu_0(z_1), \cdots, \mu_0(z_n)]$, $\kappa_0(z_{1:n}, z_{1:n}) = [\kappa_0(z_1, z_1), \cdots, \kappa_0(z_1, z_n); \cdots; \kappa_0(z_n, z_1), \cdots, \kappa_0(z_n, z_n);]$.

Now we wish to infer the value of $f(z)$ at some new point $z$, the posterior process $f(z)|\mathcal{D}_n$ is also a Gaussian process (GP) with mean $\mu_n$ and covariance $\sigma_n^2$:

$$f(z)|\mathcal{D}_n \sim \text{Normal}(\mu_n(z), \sigma_n^2(z)), \tag{2}$$
$$\mu_n(z) = \kappa_0(z, z_{1:n})\kappa_0(z_{1:n}, z_{1:n})^{-1}(f(z_{1:n}) - \mu_0(z_{1:n})) + \mu_0(z),$$
$$\sigma_n^2(z) = \kappa_0(z, z) - \kappa_0(z, z_{1:n})\kappa_0(z_{1:n}, z_{1:n})^{-1}\kappa_0(z_{1:n}, z).$$

As a optimization method to maximize a function $f$, Bayesian optimization models the function to make decisions about where to evaluate the next point $z$. Assuming we already obtained observations $\mathcal{D}_{t-1} = \{(z_i, f(z_i))\}_{i=1}^{t-1}$, to determine the next point $z_t$ for evaluation, we first use the posterior GP to define an *acquisition function* $\varphi_t : \mathcal{Z} \to \mathbb{R}$, which models the utility of evaluating $f(z)$ for any $z \in \mathcal{Z}$. We then evaluate $f(z_t)$ with

$$z_t = \arg\max_{\mathcal{Z}} \varphi_t(z). \tag{3}$$

In this work, we use the expected improvement (EI) acquisition function (Mockus et al., 1978)

$$\varphi_t(z) = \sqrt{\sigma_n^2(z)}(\gamma(z)\Phi(\gamma(z)) + \phi(\gamma(z))) \qquad \text{with} \qquad \gamma(z) = \frac{\mu_n(z) - f(z_{best})}{\sqrt{\sigma_n^2(z)}}, \tag{4}$$

which measures the expected improvement over the current best value $z_{best} = \arg\max_{z_i} f(z_i)$ according to the posterior GP. Here $\Phi(\cdot)$ and $\phi(\cdot)$ are the cdf and pdf of $\mathcal{N}(0, I)$ respectively.

## 4 SCORE-BASED BLACK-BOX ATTACK

Suppose a classifier $F(x)$ has input $x$ and label $y$. An un-targeted adversarial example $x_{adv}$ satisfies:

$$\arg\max_{j \in \{1, \cdots C\}} F(x_{adv})_j \neq y \qquad \text{and} \qquad \|x_{adv} - x\|_p \leq \varepsilon, \tag{5}$$

where $C$ is the number of classes. While an adversarial example for targeted attack means the maximum position of $F(x)$ should be the targeted class $q$: $\arg\max_{j \in \{1, \cdots C\}} F(x_{adv})_j = q$. In order to find $x_{adv}$, we may optimize a surrogate loss function $\ell(x, y)$ (e.g hinge loss).

In this work, we consider adversarial attack as a time-varying contextual bandits problem. At each time $t$, we observe a state $x_t$ which is a modification of the original input $x_0$. Before taking arm $a_t \in \mathcal{A} \subset \mathbb{R}^d$, we could observe the feature $z$ of arms. And $a_t$ would modify state $x_t$ to $x_{t+1}$ according to

$$x_{t+1} = \arg\min_{s \in \{x_t + a_t, x_t\}} \ell(\Pi_{B_p(x, \varepsilon)}(s), y) \tag{6}$$

with reward function $r(x_t, a_t) = \ell(x_{t+1}, y) - \ell(x_t, y)$ and the checking step tries to remove negative reward. In this frame, we would like to estimate the reward $r(x_t, a_t)$ with feature $z_t$ using learning, and then pick $a_t$ to maximize the reward. Observe that

$$r(x_t, a_t) \approx \nabla_x \ell(x_t, y)^\top (x_{t+1} - x_t), \tag{7}$$

where the gradient $\nabla_{x_t} \ell(x_t, y)$ is unknown. It follows from the formulation that we may rewrite $r(x_t, a_t)$ as a function

$$r(x_t, a_t) \approx f(x_t, x_{t+1} - x_t).$$

Since in general, we make small steps from one iteration to the next iteration, $\delta_t(a_t) = x_{t+1} - x_t$ is small. We may approximate the reward with fixed gradient locally with

$$f(x_t, \delta_t) = \tilde{f}_t(a_t),$$

We may consider the learning of reward as a time-varying contextual bandits problem with reward function $\tilde{f}_t(a_t)$ for arm $a_t$ at time $t$. Since $x_{t+1} - x_t$ is small, this time-varying bandits has slow-varying property: the function $\tilde{f}_t$ changes slowly from time $t$ to time $t + 1$.

In the proposed framework, our goal is to learn the time-varying bandits reward $\tilde{f}_t(a_t)$ with feature $z_t$. We use Gaussian process regression to model the reward function using recent historic data since the reward function is slow-varying, and describe the details in the subsequent sections.

We note that the most general action space contains all $a_t \in \mathbb{R}^d$, where $d$ is the number of image pixels. However, it is impossible to explore the arms in such a large space. In this work, we choose a specific class of actions $\mathcal{A} = \{a_i\}_{i=1}^n$, $n$ is the image blocks of different sizes. It covers the space of the adversarial perturbations while maintaining good complexity. We also find the location and the PCA of the blocks a good component of the feature $z$ associated with the arm. Besides, modifying a block only affects the state locally. Therefore the reward function remains similar after state changes.

## 4.1 Structured Search with Gaussian Process Regression and Bayesian Optimization

Define the block size as $b$, we divide the image into several blocks $E = \{e_{000}, e_{001}, \cdots, e_{hwc}\}$, where the block is $b \times b$ square of pixels and $(h, w, c) = (\text{height}/b, \text{width}/b, \text{channel})$. Each block $e_{ijk}$ is associated with the feature $z_{e_{ijk}}$ such as the location of the block.

Suppose we have time-varying bandits with state $x_t$ and unknown reward function $\tilde{f}_t$ at time $t$. By taking the action $a_{e_{ijk}}$, we change the individual block $e_{ijk}$ of $x_t$ and get $x_{t+1}$ with reward $\tilde{f}_t(a_{e_{ijk}})$. We consider two ways of taking action $a_{e_{ijk}}$ on block $e_{ijk}$ : CorrAttack$_{\text{Diff}}$ and CorrAttack$_{\text{Flip}}$.

**Finite Difference CorrAttack$_{\text{Diff}}$:** For action $a_{e_{ijk}}$, the attacker will query $\ell(x_t + \eta e_{ijk}, y)$ and $\ell(x_t - \eta e_{ijk}, y)$, and choose

$$a_{e_{ijk}} = \underset{s \in \{\eta e_{ijk}, -\eta e_{ijk}\}}{\arg\min} \ell(x_t + s, y). \tag{8}$$

The action space $\mathcal{A} = \{a_{e_{ijk}} | e_{ijk} \in E\}$.

In our framework, the bandits problem can also be regarded as learning the conditional gradient over actions. That is, when $\eta$ is small, we try to choose action $a_t$ with

$$a_t = \underset{e_{ijk} \in E}{\arg\min} e_{ijk}^\top \nabla_{x_t} \ell(x_t, y) \tag{9}$$

which is the conditional gradient over the set of blocks.

**Discrete Approximation CorrAttack$_{\text{Flip}}$:** In general, adversarial attack with $\ell_\infty$ budget can be formulated as constrained optimization with $\|x_{adv} - x\|_\infty \le \epsilon$. However, PARSI (Moon et al., 2019) limits the space to $\{-\varepsilon, +\varepsilon\}^d$, which leads to better performance for black-box attack (Moon et al., 2019). The continuous optimization problem becomes a discrete optimization problems as follows:

$$\begin{aligned} \text{maximize } \ell(x_{adv}, y) &\implies \text{maximize } \ell(x_{adv}, y) \\ \text{subject to } \|x_{adv} - x\|_\infty \le \epsilon \quad &\text{subject to } x_{adv} - x \in \{\epsilon, -\epsilon\}^d. \end{aligned} \tag{10}$$

Following PARSI, we consider two stages to perform structured search. When flipping $\varepsilon$ to $-\varepsilon$, $a_{e_{ijk}}$ changes the block to $-\varepsilon$ and $\mathcal{A} = \{-2\varepsilon e_{ijk} | e_{ijk} \in E\}$. When changing $-\varepsilon$ to $\varepsilon$, $\mathcal{A} = \{2\varepsilon e_{ijk} | e_{ijk} \in E\}$ instead.

**Gaussian Process (GP) Regression:** We model the **difference function**

$$g_t(a_t) = \ell(\Pi_{B_p(x,\varepsilon)}(x_t + a_t), y) - \ell(x_t, y) \tag{11}$$

instead of the reward function $\tilde{f}_t(a_t) \geq 0$, as the difference function could be negative, providing more information about the negative arms in $\mathcal{A}$. We would collect historic actions with feature and difference $\{z_k, g_k(a_k))\}_{k=1}^{t}$ and learn the difference to make choices at a later stage. At each time $t$, we use the Gaussian process regression to model the correlation between the features $z_{e_{ijk}}$ and use Bayesian optimization to select the next action. More specifically, the same as eq. (2), we let

$$g_t(a_{e_{ijk}})|\mathcal{D}_t \sim \text{Normal}(\mu_t(z_{e_{ijk}}), \sigma_t^2(z_{e_{ijk}})), \tag{12}$$

where $\mathcal{D}_t = \{z_k, g_k(a_k))\}_{k=t-\tau}^{t}$ is the difference of evaluated blocks $e_{t-\tau:t}$ with feature $z_{e_{t-\tau:t}}$ and $\tau$ is a parameter to forget old samples. Then we use EI acquisition function to pick up the next action $a_{t+1}$ in $\mathcal{A}$. More specifically, the same as eq. (4), we let

$$a_{t+1} = \arg\max_{\mathcal{A}}(\sqrt{\sigma_t^2(z_{e_{ijk}})}(\gamma(z_{e_{ijk}})\Phi(\gamma(z_{e_{ijk}})) + \phi(\gamma(z_{e_{ijk}})))) \tag{13}$$

As the difference function $g_t$ is varying, we take two strategies in Algorithm 2 to update the previous samples to make sure GP regression learns the current difference function well. The first strategy is to remove old samples in $\mathcal{D}_t$. Even if the bandits are slowly varying, the difference function will change significantly after a significant number of rounds. Therefore, we need to forget samples before $t - \tau$. The second strategy is to remove samples near the last block $e_{i_t j_t k_t}$ in $\mathcal{D}_t$. As we discuss later, the difference function may change significantly in a local region near the last selected block. Therefore previous samples in this local region will be inaccurate. The resulting algorithm for CorrAttack is shown in Algorithm 1, which mainly follows standard procedure of Bayesian optimization.

---

**Algorithm 1** CorrAttack

**Require:** Loss function $\ell(\cdot, \cdot)$, Input $x_0$ and its label $y$, Action space $\mathcal{A} = \{a_{e_{ijk}}|e_{ijk} \in E\}$, Parameter $c, \tau,$
    $\alpha$
  1:  Build set $\mathcal{D}_0 = \{(z_{e_{i_p j_p k_p}}, g_0(a_{e_{i_p j_p k_p}}))\}_{p=1}^{m}$ using latin hypercube sampling from $\mathcal{A}$
  2:  **repeat**
  3:     Fit the parameter of $\text{Normal}(\mu_t(z_{e_{ijk}}), \sigma_t^2(z_{e_{ijk}}))$ on $\mathcal{D}_t$ according to Equation (12)
  4:     Calculate *acquisition function* $\varphi_t(z_{e_{ijk}})$ and according to Equation (13)
  5:     Select $a_{e_{i_t j_t k_t}} = \arg\max_{\mathcal{A}} \varphi_t(z_{e_{ijk}})$ according to Equation (13)
  6:     $x_{t+1} = \arg\min_{s \in \{x_t + a_{e_{i_t j_t k_t}}, x_t\}} \ell(\Pi_{B_p(x,\varepsilon)}(s), y)$
  7:     Update sample set $\mathcal{D}_t$ with Algorithm 2
       $\mathcal{D}_{t+1} = \text{UPDATESAMPLES}(\mathcal{D}_t, x_t, x_{t+1}, e_{i_t j_t k_t}, g_{t+1}(a_{e_{i_t j_t k_t}}), \tau, \alpha)$
  8:  **until** $\max_{\mathcal{A}} \varphi_t(z_{e_{ijk}}) < c$
  9:  **return** $x_T$;

---

**Algorithm 2** Update Samples

**Require:** Sample set $\mathcal{D}_t$, State $x_t, x_{t+1}$, Block $e_{i_t j_t k_t}$, Difference $g_{t+1}(a_{e_{i_t j_t k_t}})$, Paramter $\tau, \alpha$
  1:  **if** $x_{t+1} \neq x_t$ **then**
  2:     $\mathcal{D}_{t+1} = \mathcal{D}_t \setminus \{(z_{e_{ijk}}, g) \in \mathcal{D}_t || i - i_t| + |j - j_t| \leq \alpha\}$
  3:  **else**
  4:     $\mathcal{D}_{t+1} = \mathcal{D}_t \cup \{(z_{e_{i_t j_t k_t}}, g_{t+1}(a_{e_{i_t j_t k_t}}))\}$
  5:  **end if**
  6:  Remove the earliest sample from $\mathcal{D}$ if the cardinality $|\mathcal{D}| > \tau$
  7:  **return** $\mathcal{D}_{t+1}$

---

### 4.2   Features and Slow Varying Property

**Features of Contextual Bandits:** We use a four dimensional vector as the feature $z_{e_{ijk}}$:

$$z_{e_{ijk}} = (i, j, k, pca) \tag{14}$$

where $i, j, k$ is the location of the block. And $pca$ is the first component of PCA decomposition of $[x_0(e_{000}), x_0(e_{001}), \cdots x_0(e_{hwc})]$. $x_0(e_{ijk})$ means the block of natural image at the given position.

The reward function depends on the gradient in Equation (7). It has been shown that the gradient $\nabla_x \ell(x, y)$ has local dependencies (Ilyas et al., 2019). Suppose two coordinates $e_{ijk}$ and $e_{lpq}$ are close, then $\nabla_x \ell(x, y)_{ijk} \approx \nabla_x \ell(x, y)_{lpq}$. We consider the finite difference of the block $e_{ijk}$

$$\Delta_t(e_{ijk}) = \ell(x_t + \eta e_{ijk}, y) - \ell(x_t - \eta e_{ijk}, y) \approx 2\eta e_{ijk}^\top \nabla_{x_t} \ell(x_t, y) \tag{15}$$

where $\eta$ is a small step size. When $\eta$ is small, the reward can be approximated by the average of the gradients around a small region, which also has local dependencies. In fact, the local structure of the reward will also be perseveed when the block size and $\eta$ is large. Figure 1 shows one example of the finite difference $\Delta_t(e_{ijk})$ obtained on ImageNet dataset with ResNet50. This shows blocks with closer locations are more likely to have similar reward. Therefore, we add the location of the block as the feature so that it uses historic data to find the arm with the largest reward.

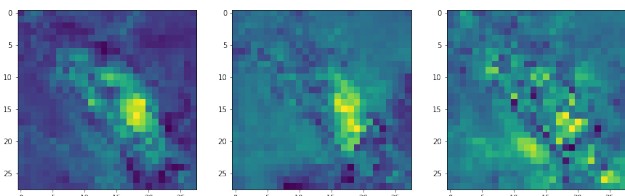

Figure 1: Finite difference of the perturbation for three channels on one image from ImageNet with ResNet50. $h = w = 28$, $b = 8$ and $\eta = 0.05$. Lighter block means larger finite difference.

In addition to the location of the difference, we may add other features. The block of the image itself forms a strong feature for the regression, but the dimension of the block is too high for GP regression. Therefore, we use PCA to lower the dimension and add the first component into the feature vector.

**Slow Varying Property** In addition to the local dependencies of finite difference, the difference would also be slow varying if we just change a small region of $x_t$. Let $x_{t+1} = x_t - \eta e_{i_t j_t k_t}$, Figure 2 shows the difference of $\Delta_t(e_{ijk})$ and $\Delta_{t+1}(e_{ijk})$, which is centralized in a small region near $e_{i_t j_t k_t}$. Reward function is based on the finite difference, which also has the slow varying property. It could be explained by the local property of convolution. When $\eta$ is small, the finite difference can be approximated with gradient and the local Hessian:

$$\Delta_{t+1}(e_{ijk}) - \Delta_t(e_{ijk}) \approx \eta^2 e_{ijk}^\top \nabla_{x_t}^2 \ell(x_t, y) e_{i_t j_t k_t} \tag{16}$$

The difference is much smaller than $\Delta_t(e_{ijk})$. Today's neural networks are built with stacks of convolutions and non-linear operations. Since these operations are localized in a small region, the Hessian of a neural network is also localized and the reward function only changes near $e_{i_t j_t k_t}$.



Figure 2: Difference of finite difference on each block after changing block $e_{15,18,1}$ of Figure 1, which is the lightest pixel in the picture. Darker blocks imply smaller difference in finite difference, which is almost zero in the majority of the image except the part near the changed block.

### 4.3 HIERARCHICAL BAYESIAN OPTIMIZATION SEARCH

Recent black-box approaches (Chen et al., 2017; Moon et al., 2019) exploit the hierarchical image structure for query efficiency. Following these approaches, we take a hierarchical approach and perform the accelerated local search in Algorithm 1 from a coarse grid (large blocks) to a fine grid (smaller blocks). The algorithm for hierarchical attack iteratively performs Algorithm 1 at one block size, and then divides the blocks into smaller sizes. At each block size, we build a Gaussian process to model the difference function, and perform structured search with the blocks until $\max_{\mathcal{A}} \varphi_t(z_{e_{ijk}}) < c$. When dividing the blocks into smaller sizes, we will build a new block set $E$ with action $a_{e_{ijk}}$ and new feature $z_{e_{ijk}}$, but keep the $x_t$ in last block size as $x_0$ in new block size. Define the stage as $\mathcal{S} = \{0, 1, \cdots, s\}$ and initial block size as $b$. The block at stage $s$ is $\frac{b}{2^s} \times \frac{b}{2^s}$ square of pixels and $(h, w, c) = (2^s * \text{height}/b, 2^s * \text{width}/b, \text{channel})$.

The overall hierarchical accelerated local search algorithm is shown in Appendix A. It is important to note that most of the attacks terminate in the early stages and rarely need to run on fine scales.

Table 1: Success rate and average queries of un-targeted attack on 1000 samples of ImageNet. $\varepsilon = 0.05$. Since BayesOpt and Bayes-Attack needs thousands of hours to run all samples, we only test 20 samples, which are marked as *, the complexity and running time could be referred to C.6.

| Attack | VGG16 | | Resnet50 | | Densenet121 | |
|---|---|---|---|---|---|---|
| | Success | Queries | Success | Queries | Success | Queries |
| ZOO | 81.93% | 2003 | 63.68% | 1795 | 76.73% | 1864 |
| NES | 99.72% | 700 | 99.19% | 1178 | 99.72% | 1074 |
| NAttack | **100%** | 293 | 99.73% | 401 | **100%** | 375 |
| Bandits | 94.75% | 389 | 96.92% | 433 | 98.09% | 635 |
| PARSI | **100%** | 365 | 99.73% | 432 | **100%** | 387 |
| Square Attack | **100%** | **79** | **100%** | **112** | **100%** | **86** |
| SignHunter | **100%** | 104 | **100%** | 145 | **100%** | 118 |
| CorrAttack$_{\text{Diff}}$ | **100%** | 389 | 99.86% | 419 | 99.86% | 334 |
| CorrAttack$_{\text{Flip}}$ | **100%** | 130 | **100%** | 150 | **100%** | 113 |
| BayesOpt* | **100%** | 182 | **100%** | 214 | **100%** | 223 |
| Bayes-Attack* | **100%** | 244 | **100%** | 254 | **100%** | 213 |
| CorrAttack$_{\text{Flip}}$* | **100%** | **110** | **100%** | **96** | **100%** | **87** |

Table 2: Success rate and average queries of targeted attack on ImageNet. $\varepsilon = 0.05$ and query limit is 10000. As BayesOpt and Bayes-Attack run very slow, we do not include them for the targeted attack.

| Attack | VGG16 | | Resnet50 | | Densenet121 | |
|---|---|---|---|---|---|---|
| | Success | Queries | Success | Queries | Success | Queries |
| ZOO | 1.1% | 2884 | 0.8% | 3018 | 1.1% | 3309 |
| NES | 80.82% | 4582 | 52.73% | 5762 | 64.21% | 5427 |
| NAttack | 91.86% | 4045 | 89.05% | 3799 | 91.97% | 4389 |
| Bandits | 50.62% | 5379 | 40.18% | 5672 | 43.53% | 5434 |
| PARSI | 76.28% | 3229 | 64.88% | 3403 | 75.09% | 3246 |
| Square Attack | 96.69% | **2060** | 89.52% | 2807 | 95.38% | 2280 |
| SignHunter | 93.52% | 2999 | 83.71% | 3905 | 90.75% | 3632 |
| CorrAttack$_{\text{Diff}}$ | 88.41% | 3826 | 81.84% | 4064 | 91.29% | 3513 |
| CorrAttack$_{\text{Flip}}$ | **98.07%** | 2191 | **96.39%** | **2531** | **99.41%** | **2019** |

## 5 EXPERIMENTS

We evaluated the number of queries versus the success rates of CorrAttack on both undefended and defended network on ImageNet (Russakovsky et al., 2015). Moreover, we attacked Google Cloud Vision API to show that CorrAttack can generalize to a true black-box model.

We used the common hinge loss proposed in the CW attack (Carlini & Wagner, 2017). We compared two versions of CorrAttack : CorrAttack$_{\text{Diff}}$ and CorrAttack$_{\text{Flip}}$, to ZOO (Chen et al., 2017), NES (Ilyas et al., 2018), NAttack (Li et al., 2019), Bandits (Ilyas et al., 2019), PARSI (Moon et al., 2019), Square Attack (Andriushchenko et al., 2020), SignHunter(Al-Dujaili & O'Reilly, 2020b), BayesOpt (Ru et al., 2020) and Bayes-Attack (Shukla et al., 2019). We only test adversarial attack on $\ell_\infty$ norm. The details of the Gaussian processes regression and the hyperparameters of CorrAttack are given in the Appendix B. We shall mention that CorrAttack is not sensitive to the hyperparameters. The hyperparameters of other methods follow those suggested by the original papers.

### 5.1 UNDEFENDED NETWORK

We randomly select 1000 images from the validation set of ImageNet and only attack correctly classified images. The query efficiency of CorrAttack is tested on VGG16 (Simonyan & Zisserman, 2014), Resnet50 (He et al., 2016) and Densenet121 (Huang et al., 2017), which are the most commonly used network structures. We set $\varepsilon = 0.05$ and the query limit to be 10000 except for BayesOpt and Bayes-Attack. For targeted attacks, we randomly choose the target class for each image and the target classes are maintained the same for the evaluation of different algorithms. The results are shown in Table 1 and 2. CorrAttack$_{\text{Flip}}$ outperforms other methods by a large margin.

Table 3: Success rate and average queries of un-targeted attack on defended model. Since BayesOpt and Bayes-Attack take thousands of hours to run, we only tested on 10 samples from ImageNet with $\varepsilon = 0.05$ and 1000 query limit, which are marked as *.

| Method | ZOO | NES | NAttack | Bandits | PARSI |
|---|---|---|---|---|---|
| Success | 28.57% | 24.13% | 74.38% | 55.82% | 73.40% |
| Queries | 1954 | 3740 | 1078 | 1873 | 1529 |
| Method | SignHunter | Square Attack | CorrAttack$_{Diff}$ | | CorrAttack$_{Flip}$ |
| Success | 68.97% | 73.89% | 64.86% | | **79.15%** |
| Queries | 1392 | 1086 | 1599 | | **1036** |
| Method | BayesAttack* | | BayesOpt* | | CorrAttack$_{Flip}$* |
| Success | 50.00% | | 50.00% | | 60.00% |
| Queries | 129 | | 406 | | 206 |

Table 4: Success rate and average queries of un-targeted attack on Google Cloud Vision API. $\varepsilon = 0.05$

| Method | NAttack | BayesOpt | PARSI | CorrAttack$_{Flip}$ |
|---|---|---|---|---|
| Success | 70.00% | 30.00% | 70.00% | 80.00% |
| Queries | 142 | 129 | 235 | 155 |

As BayesOpt and Bayes-Attack takes tens of thousands of hours to attack 1000 images, we compare them with CorrAttack$_{Flip}$ only on 20 images and un-targeted attack. The query limit is also reduced to 1000 as the time for BayesOpt and Bayes-Attack quickly increases as more samples add into the Gaussian distribution. The time comparison between three models is shown in Appendix C.6.

**Optimality of Bayesian optimization** Appendix C.1 shows the rank the actions found by CorrAttack. The attacker could find the action with large reward quickly.

**Varying $\varepsilon$** We also test the algorithms at different budget of adversarial perturbations at $\varepsilon = 0.04$ and $\varepsilon = 0.06$ on Resnet50. As it is shown in Appendix C.2, CorrAttack shows a consistently better performance at different $\varepsilon$.

**Ablation study on random choices** Appendix C.3 shows the ablation study of random version of CorrAttack$_{Diff}$ and CorrAttack$_{Flip}$. In both cases, Bayesian optimization helps to gain better query efficiency.

**Ablation study on hierarchical attack** We perform ablation study on the hierarchical attack and the result is shown in Appendix C.4. Hierarchical structure accelerates the CorrAttack and eliminates the sensitivity of choosing initial block size.

**Ablation study on features** Appendix C.5 demonstrates how the feature of the contextual bandits affects the performance of attack. PCA would help to improve the efficiency of attack.

## 5.2 DEFENDED NETWORK

To evaluate the effectiveness of CorrAttack on adversarially defended networks, we tested our method on one of the SOTA robust model (Xie et al., 2018) on ImageNet. The weight is downloaded from Github[1]. "ResneXt DenoiseAll" is chosen as the target model as it achieves the best performance. We set $\varepsilon = 0.05$ and the maximum number of queries is 10000. As BayesOpt runs very slowly, the attack is also performed on 10 images and the query limit is 1000. The result is shown in Table 3. CorrAttack$_{Flip}$ still outperforms other methods.

## 5.3 GOOGLE CLOUD VISION API

We also attacked Google Cloud Vision API, a real world black-box model for classification. The target is to remove the top-1 label out of the classification output. We choose 10 images for the ImageNet dataset and set the query limit to be 500 due to high cost to use the API. We compare CorrAttack$_{Flip}$ with NAttack, BayesOpt and PARSI. The result is shown in Table 4. We also show one example of the classification output in Appendix C.9

---

[1]https://github.com/facebookresearch/ImageNet-Adversarial-Training

## 6  CONCLUSION AND FUTURE WORK

We formulate the score-based adversarial attack as a time-varying contextual bandits and propose a new method CorrAttack. By performing structured search on the blocks of the image, the bandits has the slow varying property. CorrAttack takes advantage of the the features of the arm, and uses Bayesian optimization with Gaussian process regression to learn the reward function. The experiment shows that CorrAttack can quickly find the action with large reward and CorrAttack achieves superior query efficiency and success rate on ImageNet and Google Cloud Vision API.

We only include basic features for learning the bandits. Other features like embedding from the transfer-based attack Huang & Zhang (2020) may be taken into account in the future work. While our work only focuses on adversarial attack on $\ell_\infty$ norm, the same contextual bandits formulation could be generalized to other $\ell_p$ norm to improve query efficiency. Besides, defense against CorrAttack may be achieved with adversarial training on CorrAttack , but it may not be able to defend other attacks in the meantime.

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

## A   ALGORITHM

---

**Algorithm 3** Split Block

---

**Require:** Set of blocks $E$, Block size $b$, $E' = \emptyset$
1: **for** each block $e \in E$ **do**
2:     Split the block $e$ into 4 blocks $\{e_1, e_2, e_3, e_4\}$ with size $b/2$
3:     $E' \leftarrow E' \cup \{e_1, e_2, e_3, e_4\}$
4: **end for**
5: **return** $E'$;

---

---

**Algorithm 4** Hierarchical CorrAttack$_{\text{Diff}}$

---

**Require:** Loss function $\ell(\cdot, \cdot)$, Input image $x$ and its label $y$, Initial Block size $b$, Set of blocks $E$ containing all
    blocks of the image, Threshold $c, \tau, \alpha$, Step size $\eta$, Adversarial budget $\varepsilon$
1: $x_0 = x$
2: **repeat**
3:     Choose $A = \{a_{e_{ijk}} | e_{ijk} \in E\}$ with Equation (8)
4:     Run CorrAttack on current block size
        $x = \text{CORRATTACK}\,(\ell(\cdot, \cdot), x, y, A, c, \tau, \alpha)$
5:     **if** $b > 1$ **then**
6:         Split the blocks into finer blocks using Algorithm 3
            $E = \text{SPLITBLOCK}(E, b)$
7:         $b \leftarrow b/2$
8:     **end if**
9: **until** $\ell$ converges
10: **return** $x_K$;

---

---

**Algorithm 5** Hierarchical CorrAttack$_{\text{Flip}}$

---

**Require:** Loss function $\ell(\cdot, \cdot)$, Input image $x$ and its label $y$, Block size $b$, Set of blocks $E$ containing all blocks
    of the image, Threshold $c, \tau, \alpha$, Adversarial budget $\varepsilon$
1: $x_0 = x$
2: **for** $e_{ijk} \in E$ **do**
3:     Randomly draw $v$ from $\{-\varepsilon, \varepsilon\}$
4:     $x_0[e_{ijk}] = v + x_0[e_{ijk}]$
5: **end for**
6: **repeat**
7:     $A_n = \{2\varepsilon e_{ijk} \in E | e_{ijk}^\top (x_k - x) < 0\}$
8:     Run CorrAttack flipping $-\varepsilon$ to $\varepsilon$
        $\tilde{x}_k = \text{CORRATTACK}\,(\ell(\cdot, \cdot), x_k, y, A_n, c, \tau, \alpha)$
9:     $A_p = \{-2\varepsilon e_{ijk} \in E | e_{ijk}^\top (\tilde{x}_k - x) > 0\}$
10:    Run CorrAttack flipping $\varepsilon$ to $-\varepsilon$
        $x_{k+1} = \text{CORRATTACK}\,(\ell(\cdot, \cdot), \tilde{x}_k, y, A_p, c, \tau, \alpha)$
11:    **if** $b > 1$ **then**
12:        Split the blocks into finer blocks using Algorithm 3
            $E = \text{SPLITBLOCK}(E, b)$
13:        $b \leftarrow b/2$
14:    **end if**
15: **until** $\ell$ converges
16: **return** $x_K$;

---

## B   DETAILS OF EXPERIMENT SETTING

We use the hinge loss for all the experiments. For un-targeted attacks,

$$\ell_{\text{untarget}}(x, y) = \max \left\{ F(x)_y - \max_{j \neq y} F(x)_j, -\omega \right\} \tag{17}$$

and for targeted attacks,

$$\ell_{\text{target}}(x, y) = \max \left\{ \max_j F(x)_j - F(x)_t, -\omega \right\}. \tag{18}$$

Here $F$ represents the logits of the network outputs, $t$ is the target class, and $\omega$ denotes the margin. The image will be projected into the $\varepsilon$-ball. Besides, the value of the image will be clipped to range $[0, 1]$.

### B.1 GAUSSIAN PROCESS REGRESSION AND BYAESIAN OPTIMIZATION

We further provide details on both the computational scaling and modeling setup for the GP regression.

To address computational issues, we use `GPyTorch` (Gardner et al., 2018) for scalable GP regression. `GPyTorch` follows (Dong et al., 2017) to solve linear systems using the conjugate gradient (CG) method and approximates the log-determinant via the Lanczos process. Without `GPyTorch`, running BO with a GP regression for more than a few thousand evaluations would be infeasible as classical approaches to GP regression scale cubically in the number of data points.

On the modeling side, the GP is parameterized using a Matérn-$5/2$ kernel with ARD and a constant mean function for all experiments. The GP hyperparameters are fitted before proposing a new batch by optimizing the log-marginal likelihood. The domain is rescaled to $[0, 1]^d$ and the function values are standardized before fitting the GP regression. We use a Matérn-$5/2$ kernel with ARD for CorrAttack and use the following bounds for the hyperparameters: (length scale) $\lambda_i \in [0.005, 2.0]$, (output scale) $\lambda'_i \in [0.05, 20.0]$, (noise variance) $\sigma^2 \in [0.0005, 0.1]$.

### B.2 HYPERPARAMETERS

For CorrAttack in Algorithm 4 and Algorithm 5, we set the initial block size $b$ to be 32 and the step size $\eta$ for CorrAttack$_{\text{Diff}}$ is 0.03. In Algorithm 1, we use the initial sampling ratio $m = 0.03n$ at the start point for Gaussian process regression, the threshold $c = 10^{-4}$ to decide when to stop the search of current block size. In Algorithm 2, the threshold is different for different block size. For CorrAttack$_{\text{Flip}}$, $\alpha = 1, 1, 2, 2, 3$ for block size 32, 16, 8, 4, 2 and for CorrAttack$_{\text{Diff}}$, $\alpha = 0, 0, 1, 1, 2$ for block size 32, 16, 8, 4, 2. We set $\tau = 3m = 0.09n$ to remove the earliest samples from $D$ once $|D| > \alpha$. The Adam optimizer is used to optimize the mean $\mu$ and covariance $\kappa$ of Gaussian process, where the iteration is 1 and the learning rate is 0.1.

For PARSI, the block size is set to 32 as CorrAttack , other hyperparameters are the same as the original paper.

For Bandits, Bayes-Attack and BayesOpt, the hyperparameters are the same as the original paper.

We optimize the hyperparameters for ZOO, NES. For un-targeted attack on NES, we set the sample size to be 50, learning rate to be 0.1. For targeted attack on NES, the sample size is also 50 and the learning rate is 0.05. The learning is decay by 50% if the loss doesn't decrease for 20 iterations.

For NAttack, we set the hyperparameters the same as NES and add momentum and learning rate decay, which are not mentioned in the original paper.

For ZOO, we set the learning rate to 1.0 and sample size to be 50. Other setting follows the original paper.

## C  ADDITIONAL EXPERIMENTS

### C.1 OPTIMALITY OF BAYESIAN OPTIMIZATION

Figure 3 shows the reward function that the Bayesian optimization could find in the action set. CorrAttack could find the action with high reward within just a few queries. It shows that the Gaussian process regression could model the correlation of the reward function and the Bayesian optimization could use it to optimize the time-varying contextual bandits.

### C.2 VARYING THE ADVERSARIAL BUDGET

We test CorrAttack on different adversarial budget on ImageNet for both un-targeted attack and targeted attack. Table 5 and Table 6 show the success rate and average queries for $\varepsilon = 0.04, 0.05, 0.06$. CorrAttack$_{\text{Flip}}$ achieves the best performance among all methods.

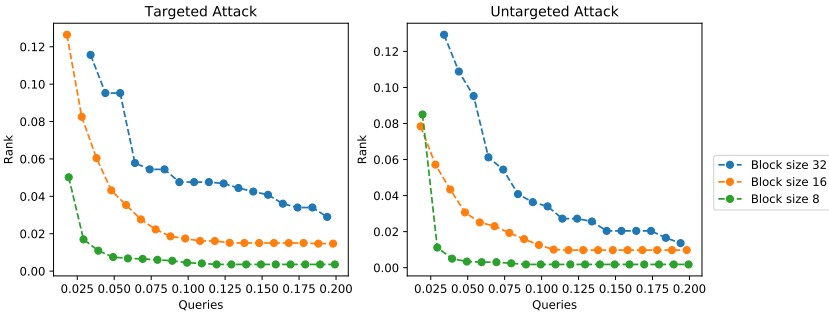

Figure 3: The rank of the reward function that the Bayesian optimization could find in the action set for different block size. The rank and query are normalized by the cardinality of the action set.

Table 5: Success rate and average queries of un-targeted attack on different $\varepsilon$. Query limit is 10000

| Attack | $\varepsilon = 0.04$ | | $\varepsilon = 0.05$ | | $\varepsilon = 0.06$ | |
|---|---|---|---|---|---|---|
| | Success | Queries | Success | Queries | Success | Queries |
| ZOO | 63.28% | 1915 | 63.68% | 1794 | 64.88% | 1507 |
| NES | 99.06% | 1230 | 99.19% | 1178 | 99.19% | 1160 |
| NAttack | 99.73% | 529 | 99.73% | 401 | 99.73% | 369 |
| Bandits | 95.86% | 898 | 96.92% | 694 | 97.06% | 567 |
| PARSI | 99.73% | 508 | 99.73% | 432 | **100%** | 387 |
| CorrAttack$_{\text{Diff}}$ | 99.86% | 479 | 99.86% | 419 | 99.78% | 373 |
| CorrAttack$_{\text{Flip}}$ | **100%** | **203** | **100%** | **150** | **100%** | **107** |

Table 6: Success rate and average queries of targeted attack on different $\varepsilon$. Query limit is 10000

| Attack | $\varepsilon = 0.04$ | | $\varepsilon = 0.05$ | | $\varepsilon = 0.06$ | |
|---|---|---|---|---|---|---|
| | Success | Queries | Success | Queries | Success | Queries |
| ZOO | 0.80% | 3514 | 0.80% | 3018 | 0.93% | 1938 |
| NES | 49.13% | 5901 | 52.73% | 5762 | 56.48% | 5884 |
| NAttack | 78.24% | 5019 | 89.05% | 3799 | 90.25% | 4321 |
| Bandits | 31.78% | 5721 | 40.19% | 5672 | 43.39% | 5609 |
| PARSI | 57.00% | 3599 | 64.88% | 3403 | 68.75% | 3250 |
| CorrAttack$_{\text{Diff}}$ | 78.70% | 4472 | 81.84% | 4064 | 85.31% | 3837 |
| CorrAttack$_{\text{Flip}}$ | **93.44%** | **2689** | **96.39%** | **2531** | **97.50%** | **2194** |

### C.3 ABLATION STUDY ON RANDOM CHOICES

Table 7 and Table 8 show the ablation study on the strategy to choose action $x_{t+1}$ in the line 6 of Algorithm 1. The process of Bayesian optimization helps to accelerate the optimization. As targeted attack is more complicated and requires larger number of queries, CorrAttack has more advantage in this scenario.

### C.4 ABLATION STUDY ON HIERARCHICAL ATTACK

We perform un-targeted attack on Resnet50 as shown in Table 10. Hierarchical attack lowers the average queries and improves the query efficiency. Besides, hierarchical attack avoids the problem of choosing block size. As shown in Table 10, block size for non-hierarchical is essential for the performance.

### C.5 ABLATION STUDY ON FEATURES

Table 11 shows the success rate and average queries for CorrAttackwith different features. We perform ablation study on the features of the contextual bandits. One contains just the location of the

Table 7: Ablation study on random choices with success rate and average queries of un-targeted attack on ImageNet. $\varepsilon = 0.05$ and query limit is 10000

| Attack | VGG16 | | Resnet50 | | Densenet121 | |
|---|---|---|---|---|---|---|
| | Success | Queries | Success | Queries | Success | Queries |
| CorrAttack$_{\text{Diff}Random}$ | **100%** | 456 | **99.86%** | 491 | **100%** | 375 |
| CorrAttack$_{\text{Diff}Bayes}$ | **100%** | **389** | **99.86%** | **419** | **100%** | **334** |
| CorrAttack$_{\text{Flip}Random}$ | **100%** | 143 | **100%** | 176 | **100%** | 132 |
| CorrAttack$_{\text{Flip}Bayes}$ | **100%** | **130** | **100%** | **150** | **100%** | **113** |

Table 8: Ablation study on random choices with success rate and average queries of targeted attack on ImageNet. $\varepsilon = 0.05$ and query limit is 10000

| Attack | VGG16 | | Resnet50 | | Densenet121 | |
|---|---|---|---|---|---|---|
| | Success | Queries | Success | Queries | Success | Queries |
| CorrAttack$_{\text{Diff}Random}$ | 83.72% | 4388 | 74.76% | 4644 | 85.30% | 4113 |
| CorrAttack$_{\text{Diff}Bayes}$ | **88.41%** | **3826** | **81.84%** | **4064** | **91.29%** | **3513** |
| CorrAttack$_{\text{Flip}Random}$ | 96.42% | 2545 | 92.92% | 3066 | 96.87% | 2556 |
| CorrAttack$_{\text{Flip}Bayes}$ | **98.07%** | **2191** | **96.39%** | **2531** | **99.41%** | **2019** |

Table 9: Ablation study on random choices with success rate and average queries of un-targeted attack on defended model ImageNet. $\varepsilon = 0.05$ and query limit is 10000

| Method | CorrAttack$_{\text{Diff}Random}$ | CorrAttack$_{\text{Diff}Bayes}$ | CorrAttack$_{\text{Flip}Random}$ | CorrAttack$_{\text{Flip}Bayes}$ |
|---|---|---|---|---|
| Success | 57.47% | **64.86%** | 71.42% | **79.15%** |
| Queries | 1645 | **1599** | 1159 | **1036** |

Table 10: Success rate and average queries of un-targeted attack on Resnet50 for Hierarchical Strategy.

| Method | Fixed Size 4 | Fixed Size 8 | Fixed Size 16 | Fixed Size 32 | Hierarchical |
|---|---|---|---|---|---|
| Success | 100% | 100.0% | 99.47% | 89.32% | **100%** |
| Queries | 763 | 351 | 168 | **96** | 150 |

block and the other contains both the location and the PCA feature. PCA helps the learning process of the reward and achieve higher success rate and lower number of queries. PCA feature achieves significant improvement on CorrAttack$_{\text{Flip}}$. We may find more useful features in the future.

## C.6 COMPARISON BETWEEN CORRATTACK$_{\text{FLIP}}$, BAYESOPT AND BAYES-ATTACK

The main difference between BayesOpt and Bayes-Attack is using different types of GP regression (Standard GP for Bayes-Attack and Additive GP for BayesOpt), so we will consider these two models as a group when comparing with our model CorrAttack.

**Difference between CorrAttack, BayesOpt and Bayes-Attack:** For $l_\infty$ attacks, assume there are no hierarchical structure, we have blocks $E = \{e_{000}, e_{001}, \cdots, e_{hwc}\}$, where the block is $b \times b$ square of pixels and $(h, w, c) = (\text{height}/b, \text{width}/b, \text{channel})$. CorrAttack, BayesOpt (Ru et al., 2020) and Bayes-Attack (Shukla et al., 2019) all try to search the adversarial noise on $E$ with perturbation $\boldsymbol{\delta} \in [-\epsilon, \epsilon]^d$ where $d = h \times w \times c$, the perturbation of block $e_{ijk}$ at time t is $\delta^t_{e_{ijk}}$.

BayesOpt and Bayes-Attack use a GP regression directly on $\boldsymbol{\delta} \in [-\epsilon, \epsilon]^d$ (all blocks),

$$f(\boldsymbol{\delta})|\mathcal{D}_n \sim \text{Normal}(\mu_n(\boldsymbol{\delta}), \sigma^2_n(\boldsymbol{\delta})). \tag{19}$$

CorrAttack define an action space $A$ and use a standard GP regression on features $z_{e_{ijk}} = (i, j, k, pca)$ (single block),

$$g_t(a_{e_{ijk}})|\mathcal{D}_t \sim \text{Normal}(\mu_t(z_{e_{ijk}}), \sigma^2_t(z_{e_{ijk}})). \tag{20}$$

Table 11: Ablation study on features with success rate and average queries of targeted attack on ImageNet. $\varepsilon = 0.05$ and query limit is 10000. We use feature $z_{e_{ijk}} = (i, j, k, pca)$ for CorrAttack$_{\text{Diff}w\ pca}$ and CorrAttack$_{\text{Flip}w\ pca}$, use $z_{e_{ijk}} = (i, j, k)$ for CorrAttack$_{\text{Diff}w/o\ pca}$ and CorrAttack$_{\text{Flip}w/o\ pca}$.

| Attack | VGG16 | | Resnet50 | | Densenet121 | |
|---|---|---|---|---|---|---|
| | Success | Queries | Success | Queries | Success | Queries |
| CorrAttack$_{\text{Diff}w/o\ pca}$ | **88.69%** | 3892 | 81.71% | 4066 | 90.88% | 3540 |
| CorrAttack$_{\text{Diff}w\ pca}$ | 88.41% | **3826** | **81.84%** | **4064** | **91.29%** | **3513** |
| CorrAttack$_{\text{Flip}w/o\ pca}$ | **98.11%** | 2233 | 95.86% | 2682 | 98.10% | 2195 |
| CorrAttack$_{\text{Flip}w\ pca}$ | 98.07% | **2191** | **96.39%** | **2531** | **99.41%** | **2019** |

At each iteration, in BayesOpt and Bayes-Attack, the changes of overall perturbation is

$$\boldsymbol{\delta}_t - \boldsymbol{\delta}_{t-1} = \{\delta_{e_{000}}^t \cup \delta_{e_{001}}^t \cup \cdots \cup \delta_{e_{hwc}}^t\} - \{\delta_{e_{000}}^{t-1} \cup \delta_{e_{001}}^{t-1} \cup \cdots \cup \delta_{e_{hwc}}^{t-1}\}. \tag{21}$$

However, in CorrAttack,

$$\boldsymbol{\delta}_t - \boldsymbol{\delta}_{t-1} = \delta_{e_{ijk}}^t - \delta_{e_{ijk}}^{t-1}. \tag{22}$$

In conclusion, BayesOpt and Bayes-Attack view each block as a dimension, try to search the overall perturbation directly. CorrAttack defines a low dimension feature space, keep an overall perturbation and try to search an action on single block.

**Time complexity and running time:** The time complexity of fitting GP regression is $O(dn^2)$ where $d$ is the dimension of input and $n$ is the number of samples. And the dimension for CorrAttack ($d = 4$ for $z_{e_{ijk}} = (i, j, k, pca)$) is much smaller than BayesOpt and Bayes-Attack ($d = 6912$ if $h = w = 48, c = 3$). Moreover, we can convert the continuous search space of BayesOpt and Bayes-Attack from $[-\epsilon, \epsilon]^{6912}$ to discrete search space $E = \{e_{000}, e_{001}, \cdots, e_{hwc}\}$, whose number is only 6912, smaller search space could save the computation time of acquisition function.

We compare the running time for CorrAttack$_{\text{Flip}}$ with BayesOpt and Bayes-Attack on 20 images from ImageNet. Table 12 shows the running time for the un-targeted attack. We use PyTorch[2] to develop these two models. All experiments were conducted on a personal workstation with 28 Intel(R) Xeon(R) Gold 5120 2.20GHz CPUs, an NVIDIA GeForce RTX2080Ti 11GB GPU and 252G memory.

BayesOpt models the loss function with a very high dimensional Gaussian process. The decomposition of additive kernel also needs to be restarted several times. Even though we try to optimize the speed of BayesOpt with GPU acceleration, it is still very slow and takes hundreds of times more computational resources than CorrAttack .

Bayes-Attack could be regarded as a simpler version of BayesOpt, which does not add additive kernel. We do not evaluate it on targeted task (when query>1000) since GP inference time grows fast as evaluated query increases, e.g. For Bayes-Attack, when $150 <$query$< 200$, Time$=1.6s/query$; $800 <$query$< 1000$, Time $= 10.5s/query$. CorrAttack solves this problem with Time$=0.1s/query$ even when query reaches 10000. Since we forget the previous samples before $t - \tau$, our input sample $n$ will be smaller than $\tau$. The forgetting technique can not be applied into the Bayes-Attack and BayesOpt since they are searching the perturbation of all blocks so each sample needs to be remembered.

## C.7 GROWING CURVE OF SUCCESS RATE

The number of average queries is sometimes misleading due to the the heavy tail distribution of queries. Therefore in Figure 4, we plot the success rates at different query levels to show the detailed behaviors of different attacks. It shows that CorrAttack is much more efficient than other methods at all query levels.

---

[2]https://pytorch.org/

Table 12: Comparsion of running time between CorrAttack$_{Flip}$ and BayesOpt on un-targeted attack. "Per Query" means the average time needed to perform one query to the loss-oracle and "Per Image" denotes the average time to successfully attack an image. Since BayesOpt needs thousands of hours to run all samples, we only tested on 20 samples from ImageNet, which will be marked as *.

| Time | VGG16 | | Resnet50 | | Densenet121 | |
|---|---|---|---|---|---|---|
| | Per Query | Per Image | Per Query | Per Image | Per Query | Per Image |
| BayesOpt* | 28.94s | 5268s | 40.53s | 8673s | 39.57s | 8825s |
| Bayes-Attack* | 3.03s | 739s | 3.42s | 869s | 2.96s | 630s |
| CorrAttack$_{Flip}$* | 0.12s | 19s | 0.11s | 15s | 0.15s | 20s |

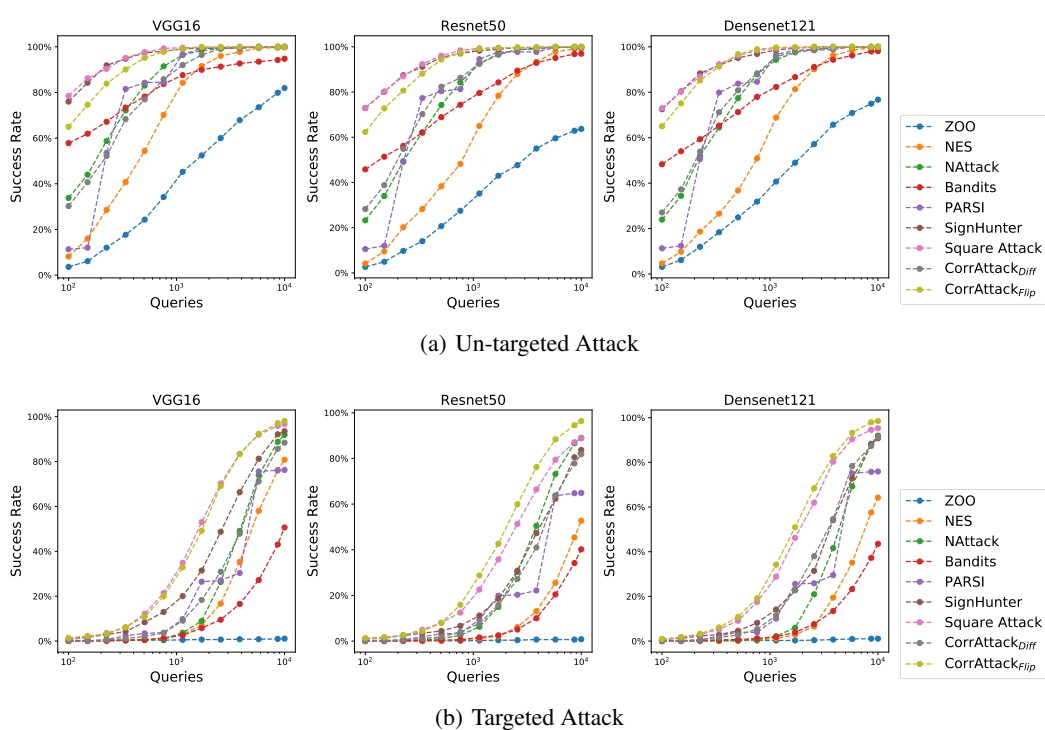

(a) Un-targeted Attack

(b) Targeted Attack

Figure 4: Success rate of black-box attack at different query levels for undefended ImageNet models.

## C.8 VISUALIZATION OF LOCAL PROPERTY AND SLOW VARYING PROPERTY

Figure 5 shows more examples of finite difference for different network architectures and different dataset. They all have local correlation structure as shown in Figure 1. And Figure 6 shows more examples like Figure 2, the slow varying properties exist for different architectures and different datasets.

## C.9 GOOGLE CLOUD VISION API

Figure 7 shows the example of attacking the Google Cloud Vision API. CorrAttack$_{Flip}$ and PARSI successfully change the classification result. BeyesOpt, however, can not remove the top 1 classification result out of the output.

## C.10 VISUALIZATION OF ADVERSARIAL EXAMPLES

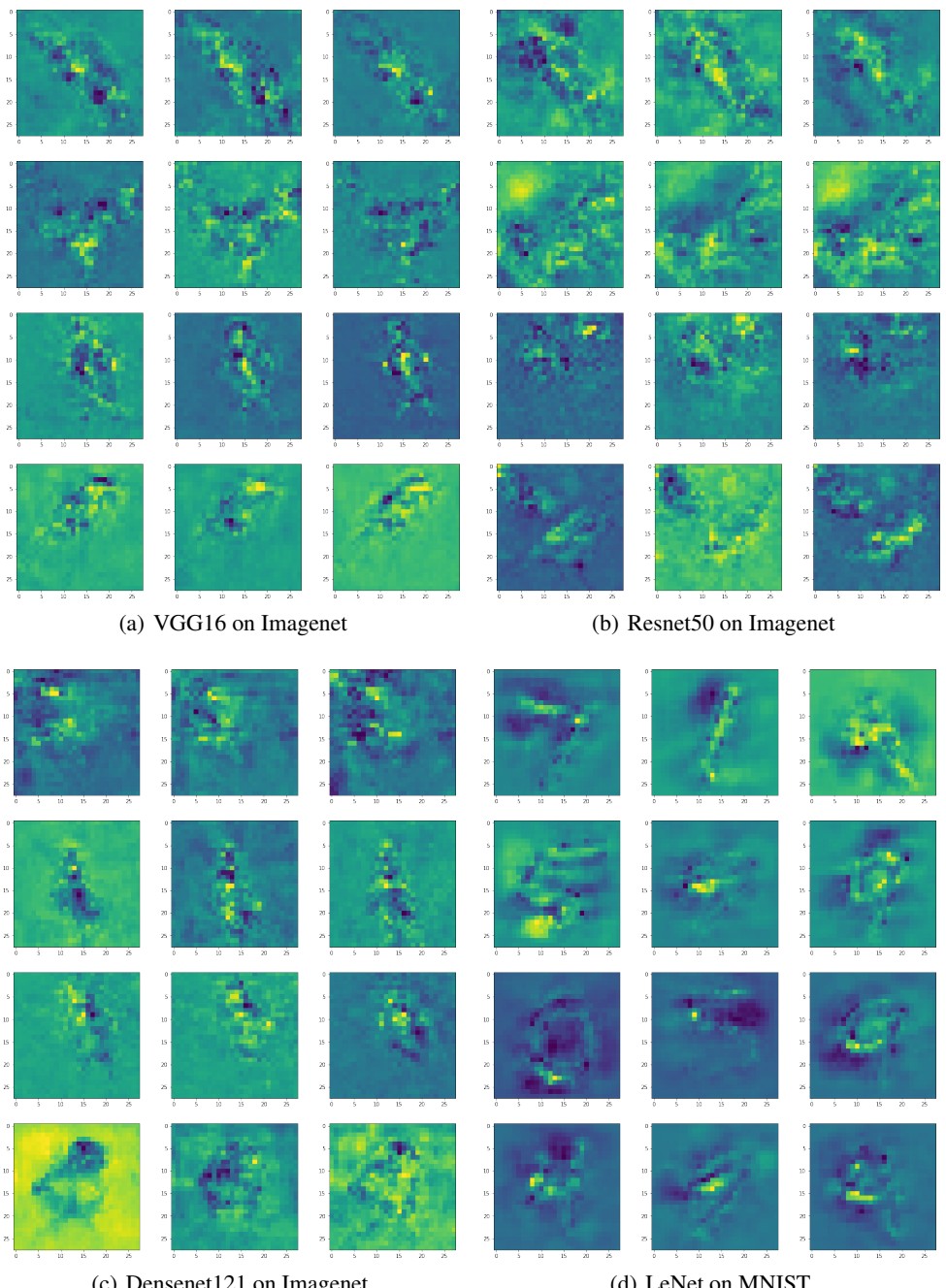

(a) VGG16 on Imagenet

(b) Resnet50 on Imagenet

(c) Densenet121 on Imagenet

(d) LeNet on MNIST

Figure 5: Finite difference of perturbation like Figure 1. $h = w = 28$. For Imagenet, $b = 8$ and $\eta = 0.05$. For MNIST, $b = 1$ and $\eta = 0.2$

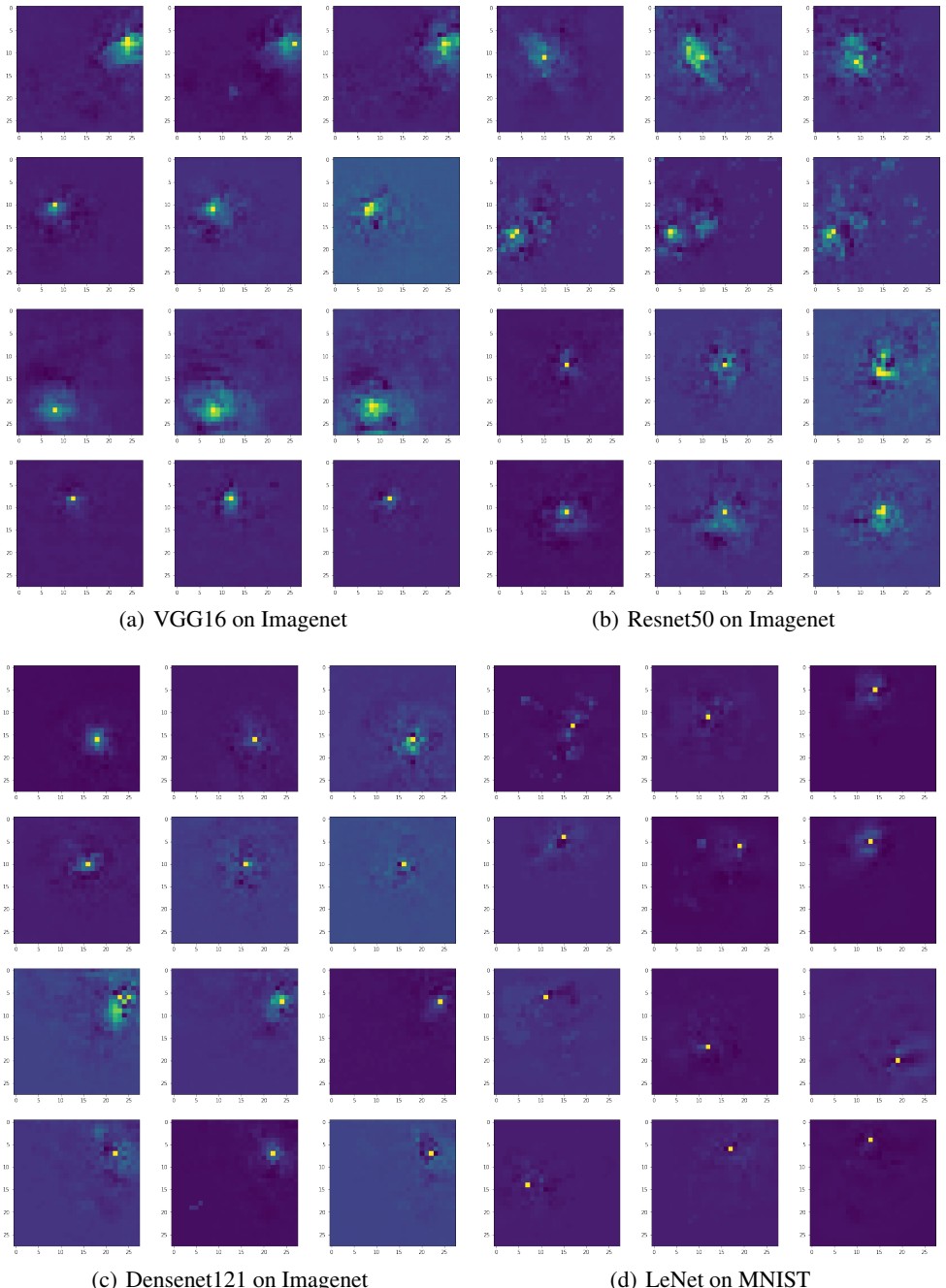

(a) VGG16 on Imagenet

(b) Resnet50 on Imagenet

(c) Densenet121 on Imagenet

(d) LeNet on MNIST

Figure 6: Difference of finite difference of perturbation like Figure 2. $h = w = 28$. For Imagenet, $b = 8$ and $\eta = 0.05$. For MNIST, $b = 1$ and $\eta = 0.2$

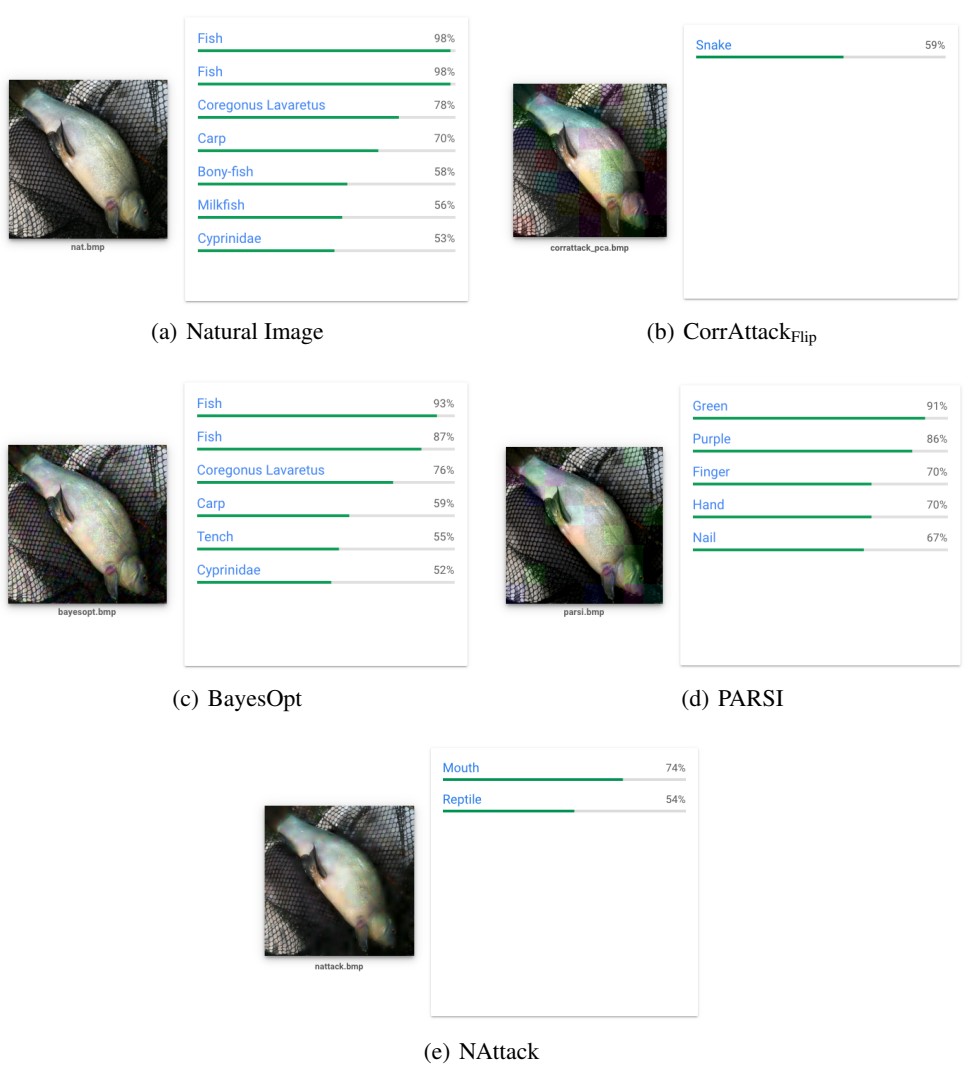

Figure 7: Example result of attacking Google Cloud Vision API

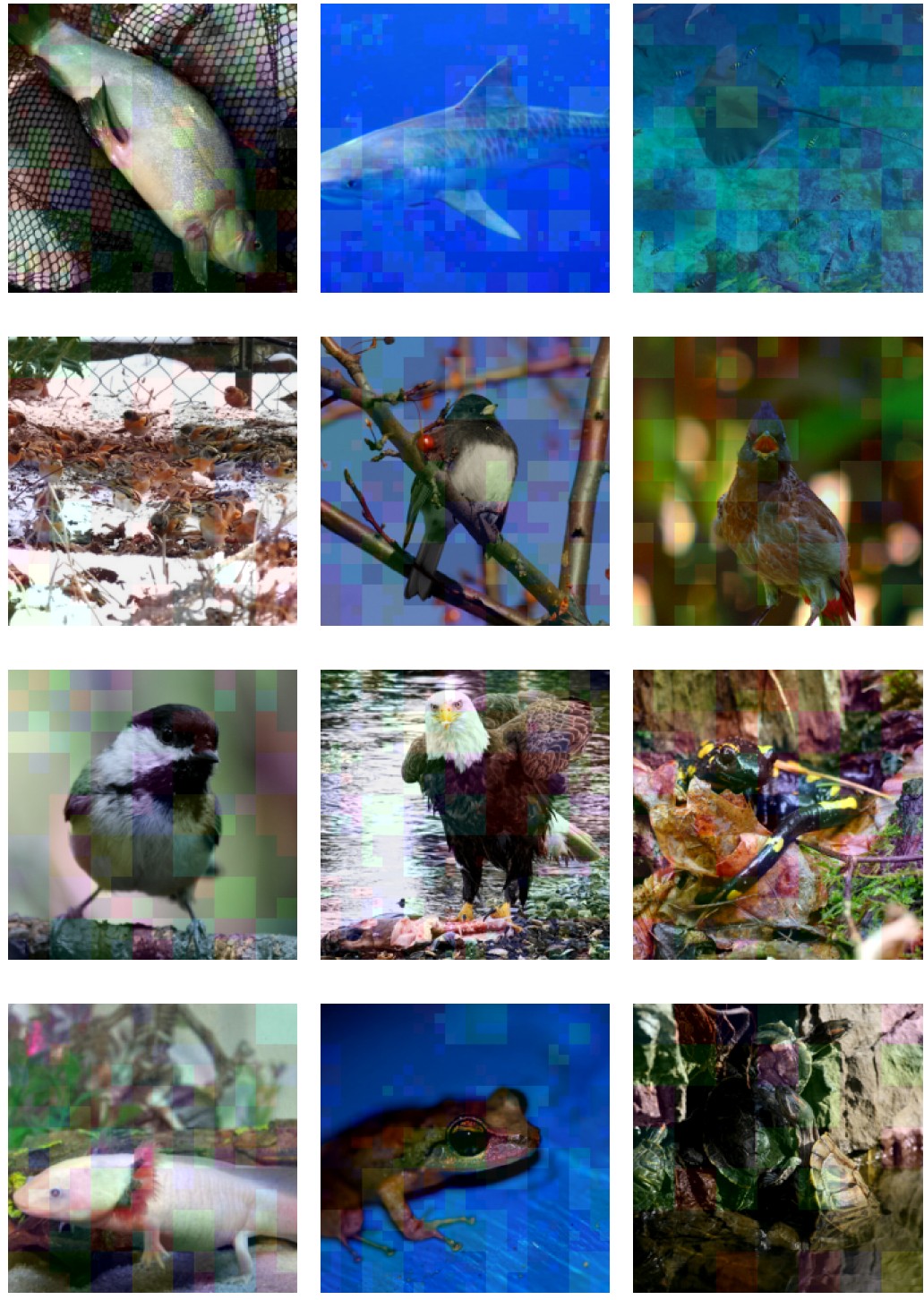

Figure 8: Visualization of adversarial examples for targeted attack on Densenet121. $\varepsilon = 0.05$

