# OpenReview forum: "CorrAttack: Black-box Adversarial Attack with Structured Search"
_ICLR.cc/2021/Conference — Reject_

### Official Review · AnonReviewer4 · 2020-10-25
**A black-box attack combining many aspects of previous attacks in one framework. Some experimental revisions are needed.**

**Rating:** 6
**Confidence:** 5

**Review:**






This paper proposes a new black-box attack that assumes access to loss-oracle of the target model. The attack exploits temporal and spatial correlation of block-wise pixel perturbations in a Bayesian optimization framework to achieve query efficiency.

Pros:
- Overall, I enjoyed reading this paper. It is one of the few papers that employ scalable GP regression in black-box attacks. It also provides a novel formulation of how the action space of perturbation should be searched by optimizing a surrogate reward function over a low-dimensional space comprising image block location and a PCA-based feature.
- The proposition combines several aspects of different attacks from the literature in one framework: sign flips, Bayesian optimization, hierarchical blocking, and exploitation of temporal- and spatial-correlation in natural images.

Cons:
- I have highlighted my remarks below and hope the authors address them.



Comments/Questions:
- Fig 1: I appreciate the authors' attempt to motivate contextual bandits and slow varying property, but making this conclusion based on one image of one dataset (ImageNet) and one architecture (ResNet50) is not sufficient. Is this observation consistent over *multiple* images of *multiple* datasets and across *multiple* architectures?
- Table 1: NAttack achieves 100% on VGG16 and DenseNet. It is not highlighted in bold.
- Table 3: query limit "1000" -> "10,000"?
- Page 2, Paragraph 2: What is the distinction between BayesOpt and Bayes-Attack. Most of the discussion is around BayesOpt.
- Page 2, Sec 2, Paragraph 2: Reference of AutoZOOM is missing.
- Page 2, Sec 2: The context in which NES and CMA-ES are presented gives the impression that they are adversarial attacks, please rephrase to indicate they are gradient-free optimization techniques.
- Eq. 9 : the notation is not clear. Is $e_{ijk}$ of the same dimensionality as $\nabla_{x_t}l(x_t,y)$?. The definition of $e_{ijk}$ does not imply that.
- It seems that most of the computational gains that CorrAttack has over BayeOpt and BayesAttack is due to the use of **scalable** GP regression (GPytorch) - not to mention the low-dimensionality of its GP. To highlight the benefits of the other aspects of CorrAttack, I think all (BayesOpt, BayesAttack, and CorrAttack) should employ the same GP tool
- Fig 3: Are the rewards across different block sizes comparable? If so, the plot suggests that using smaller blocks yields **faster** convergence to **better** rewards which in turn does not justify the hierarchical bayesian optimization search of Sec 4.3.
- PCA formulation: please elaborate on how PCA is employed to compute a single feature per block. In particular, is PCA performed for each block (pixel-level PCA) or over all the pixels of the image?
- Just as the authors compared the Bayesian aspect of the attack to Bayesian attacks (BayesOpt, Bayes-Attack), it would have been great if the authors could contrast the "flip" aspect of the attack with flip-based attacks---e.g., SignHunter & SIMBA ---that the authors mention in Sec 2.
- Ablation Studies: I appreciate the study but I think it lacks supporting evidence of "hierarchical vs non-hierarchical attack" and "small (one-pixel) block vs 32-pixel block"

---

> ### Author Response · Authors · 2020-11-24
> **Response**
>
>
> We appreciate your suggestions. We address your concerns as follows.
>
> 1. **Figure 1.** We have added more figures in the Appendix C.9 with more examples of Figure 1 and Figure 2. We choose two datasets: Imagenet and MNIST. And for Imagenet, we show the examples of three networks VGG16, Resnet50 and Densenet121.
>
> 2. **Table 1.** We have highlighted NAttack.
>
> 3. **Table 3.** Query limit is 1000 for comparison between CorrAttack-Flip, Bayes-Attack and BayesOpt. As the running time for Bayes-Attack and BayesOpt grows very fast as the number of queries grow, we are not able to compare the three methods with 10000 query limit.
>
> 4. **Difference between BayesOpt and Bayes-Attack.** BayesOpt and Bayes-Attack are similar method. One major difference is that BayesOpt uses additive Gaussian process to deal with the high dimensional problems. We have added more discussion in Appendix C.6.
>
> 5. **Reference for AutoZOOM.** We have added the reference.
>
> 6. **NES and CMA-ES.** We have rephrased the sentence.
>
> 7. **Equation 9.** $e_{ijk}$ refers to a 0-1 vector $[0,0,0,...,1,1,1,..,0,0,0]^T$ whose dimension is the same as $x_t$. Suppose block size is $b$ and $e'_{ijk}$ is a three dimensional tensor whose shape is the same as image.
>
>    $e'_{ijk}[x,y,k]= 1$ if ${b\times i \le x< b\times(i+1), b\times j \le y <b\times(j+1)}$
>
>    $e'_{ijk}[x,y,k]= 0$ otherwise
>
>    Then we get $e_{ijk}$ by reshaping $e'_{ijk}$  into a one-dimensional vector.
>
> 8. **GP speed.**  Most of the computational gains of CorrAttack is a result of the low-dimensionality of its GP. We have re-implemented CorrAttack in GPy, which is the same tool of BayesOpt and Bayes-Attack. We tested on the same 20 samples from ImageNet on un-targeted attack, CorrAttack is much faster than the other two models. We have also added more discussion in Appendix C.6 of the revision to explain the reasons.
>
>    |                 | VGG16     |           |         | Resnet50  |           |         | Densenet121 |           |         |
>    | --------------- | --------- | --------- | ------- | --------- | --------- | ------- | ----------- | --------- | ------- |
>    |                 | Per Query | Per Image | Queries | Per Query | Per Image | Queries | Per Query   | Per Image | Queries |
>    | BayesOpt        | 28.94s    | 5268s     | 182     | 40.53s    | 8673s     | 214     | 39.57s      | 8825s     | 223     |
>    | Bayes-Attack    | 3.03s     | 739s      | 244     | 3.42s     | 869s      | 254     | 2.96s       | 630s      | 213     |
>    | CorrAttack-Flip | 0.01s     | 2.07s     | 150     | 0.02s     | 2.06s     | 95      | 0.03s       | 3.46s     | 105     |
>
> 9. **Figure 3.** Figure 3 shows the convergence speed of Bayesian optimization for different block size. However, the magnitude of reward on different block size is different. Flipping block with larger size gives larger reward. As shown in the ablation study of hierarchical attack (Appendix C.4 in the revision), performing CorrAttack on small block size results in more queries than large block size.
>
> 10. **PCA.** We aggregate the pixels inside the block as a vector $v_i$. For an action space $\mathcal{A}$ with $n$ blocks, we will get a set of vectors $[v_1, v_2,..., v_{n}]$. We regard each vector $v_i$ as one instance of PCA. Then we perform PCA to transform $v_i$ to a one-dimensional vector and use it as one feature of GP.

---

> > ### Author Response · Authors · 2020-11-24
> > **Response**
> >
> >
> > 11. **SignHunter and SIMBA.** We have updated the result to Table 1 and Table2. CorrAttack outperforms SignHunter in complicated task like targeted attack and un-targeted attack of defended model.
> >
> >     SIMBA is designed for $\ell_2$ norm. The main idea of SIMBA is to randomly change the given basis. We think the only appropriate way to modify SIMBA to $\ell_\infty$ norm for fair comparison is to apply the basis of hierarchical structure to SIMBA. The resulting algorithm is a random version of CorrAttack-Diff and we denote the method as SIMBA* or CorrAttack-Diff Random. We have added the comparison in the Appendix C.3.
> >
> >     Targeted Attack
> >
> >     | Success Rate    | VGG16      | Resnet50   | Densenet121 | Average Queries | VGG16    | Resnet50 | Densenet121 |
> >     | --------------- | ---------- | ---------- | ----------- | --------------- | -------- | -------- | ----------- |
> >     | SignHunter      | 93.52%     | 83.71%     | 90.75%      | SignHunter      | 2999     | 3905     | 3632        |
> >     | SIMBA*          | 83.72%     | 74.76%     | 85.30%      | SIMBA*          | 4338     | 4644     | 4113        |
> >     | CorrAttack-Diff | 88.41%     | 81.84%     | 91.29%      | CorrAttack-Diff | 3826     | 4064     | 3513        |
> >     | CorrAttack-Flip | **98.07%** | **96.39%** | **99.41%**  | CorrAttack-Flip | **2191** | **2531** | **2019**    |
> >
> >     Un-targeted Attack
> >
> >     | Success Rate    | VGG16      | Resnet50   | Densenet121 | Average Queries | VGG16   | Resnet50 | Densenet121 |
> >     | --------------- | ---------- | ---------- | ----------- | --------------- | ------- | -------- | ----------- |
> >     | SignHunter      | **100.0%** | **100.0%** | **100.0%**  | SignHunter      | **104** | **145**  | 118         |
> >     | SIMBA*          | **100.0%** | 99.86%     | **100.0%**  | SIMBA*          | 456     | 491      | 375         |
> >     | CorrAttack-Diff | **100.0%** | 99.86%     | **100.0%**  | CorrAttack-Diff | 389     | 419      | 334         |
> >     | CorrAttack-Flip | **100.0%** | **100.0%** | **100.0%**  | CorrAttack-Flip | 130     | 150      | **113**     |
> >
> >     Un-targeted attack on the defense model.
> >
> >     |                 | SignHunter | SIMBA* | CorrAttack-Diff | CorrAttack-Flip |
> >     | --------------- | ---------- | ------ | --------------- | --------------- |
> >     | Success Rate    | 68.97%     | 57.47% | 64.86%          | **79.15%**      |
> >     | Average Queries | 1392       | 1645   | 1599            | **1036**        |
> >
> > 12. **Ablation study on hierarchical attack.** We show the result of CorrAttack-Flip with fixed block size at 4,8,16,32 for un-targeted attack on Resnet50 and update it the to Appendix C.4. They are all inferior the to the hierarchical attack. We are not able to perform the ablation study on one-pixel block as the Gaussian process on $224\times224\times3=150528$ points is slow even if the dimension of Gaussian kernel is 4. However, the trend on small block size already shows that the attack at one-pixel block will need thousands of queries.
> >
> >     |                 | Fixed Block Size 4 | Fixed Block Size 8 | Fixed Block Size 16 | Fixed Block Size 32 | Hierarchical Block Size |
> >     | --------------- | ------------------ | ------------------ | ------------------- | ------------------- | ----------------------- |
> >     | Success Rate    | 100%               | 100.0%             | 99.47%              | 89.32%              | **100%**                |
> >     | Average Queries | 763                | 351                | 168                 | **96**              | 150                     |

---

### Official Review · AnonReviewer1 · 2020-10-28
**Improves over existing methods in terms of query efficiency - not clear to me yet by how much**

**Rating:** 6
**Confidence:** 2

**Review:**

The paper proposes to use time-varying contextual bandits in order to improve the query efficiency of score-based adversarial black-box attacks.The effectiveness is demonstrated on various classifiers for ImageNet and compared against several baseline methods.

The improvements over the baseline methods seem significant, although I have one question: in Algorithm 1, what is the actual value for m used in the experiments? The appendix indicates that m=0.03n, however, the value for n isn’t given as far as I can see. This is an important detail because it determines the number of queries that CorrAttack needs to perform upfront. Is it correct that, depending the actual value of m, some of the baseline methods may actually be more efficient than CorrAttack if the attacker only wants to compute a single adversarial example?

Another question: the original ZOO paper (Chen et al., 2017) reports more than 90% attack success rate with a maximum of 1,500 queries per sample against Inception-v3, which is higher success rate and lower number of queries than reported in this paper. Did you use the same hyper parameters as in the original work?

Minor observations:
p.2: “searches over a much limited action space models the reward” -> something is missing here
p.2: “the substitute model requires a lot of training data” -> this is a bit casual
p.2: “the dimension of the embedding space is still in the thousands” -> this depends on the particular model under consideration; suggest to generalise this statement
p.3.: “score-based attack” -> “score-based attacks”
p.4.: last paragraph before Section 4.1: the specific context of images ia introduced a bit abruptly; it will be good to make more clear which parts of the paper apply to classifiers in general, and which ones only to the image domain.
p.4: “d is the image pixels” -> “d is the number of image pixels”
p.4.: “we also find that the” -> “that” should be omitted
p.5: what is “EI”?
p.8: “find the action the large award” -> something’s missing in this sentence
p.8: last paragraph: what is meant by the “embedding from the transfer-based attack”? The very last sentence about adversarial training doesn’t make much sense to me; in particular, defending against white-box attacks should also work against CorrAttack; on the other hand, the objective of adversarial training usually isn’t to defend just against one specific attack, but to make the model robust against worst-case examples within e.g. an l-p-ball, regardless of what attack algorithm may be used to construct such examples.

---

> ### Author Response · Authors · 2020-11-24
> **Response**
>
>
> Thanks for the comments. Here are the clarifications of your questions.
>
> 1. **Choice of n.** $n$ is the cardinality of the action space $\mathcal{A}$ mentioned in the first line of Algorithm 1. For example, if we perform CorrAttack-Flip and flip the "positive" blocks to the "negative" blocks, $n$ is the number of "positive" blocks of the current noise. Our method is not sensitive to the choice of $m$ as long as $m$ is not greatly changed.
>
> 2. **ZOO.** In the Table 2 of the ZOO paper, it uses 1500 iterations instead of 1500 queries to successfully attack Inception-V3. At each iteration, it performs 128 times of gradient estimation. In total, it uses 1500$\times$128=192000 queries to attack Inception-V3. Moreover, the setting of Table 2 in the ZOO paper is different from Table 1 in our paper. In the Table 2 of the ZOO paper, it achieves small L2 distortion at 1.19, and the constraint is much tighter than ours. In conclusion, the result in their paper is not comparable to the result in our paper.
>
>    We use the same hyperparameter as the original ZOO paper except for the batch size. We set the batch size to 50 as we find it slightly improves the performance of ZOO. For example, when performing un-targeted attack on Resnet50, the success rate is improved 63.41% from to 63.68%. And the average queries is improved from 1965 to 1795.
>
> 3. **Minor observations.** We will update the paper according to your advice. And here are some clarification.
>
>    1. **“the dimension of the embedding space is still in the thousands”.** We refer to the embedding space of BayesOpt and Bayes-Attack. We will make the statement more rigorous.
>    2. **"EI".** EI is the expected improvement for calculating the acquisition function. It is introduced in the section 3.
>    3. **“embedding from the transfer-based attack”.** We mean the embedding space like [TREMBA (ICLR 2020)](https://openreview.net/forum?id=SJxhNTNYwB) for black-box attack. We will add reference here.
>    4. **Adversarial Training.** We agree with the reviewer that the adversarial training should defend all potential attacks instead of one attack. But if someone only hopes to defend some efficient black-box attacks such as CorrAttack, taking these perturbations into adversarial training may significantly improve its performance against the specific attack.

---

### Official Review · AnonReviewer2 · 2020-10-29
**Interesting modeling approach but some competing approaches are omitted and it's unclear how much Bayesian Optimization really helps**

**Rating:** 6
**Confidence:** 4

**Review:**

**Summary:**
The paper proposes a new approach for generating black-box adversarial attacks based on slowly-varying contextual bandits. The proposed approach achieved query efficiency and success rate which are not too far away from the current state of the art.

**Pros:**
- High query efficiency and success rate (although not state-of-the-art, see Cons below).
- Better runtime than other Bayesian optimization methods (but it’s unclear to me how effective the Bayesian optimization approach is in the first place, see Cons below).
- The paper is clearly written.

**Cons:**
- My main concern is the list of competitors in Table 1 which does not include [SignHunter (ICLR’20)](https://openreview.net/pdf?id=SygW0TEFwH) or [Square Attack (ECCV’20)](https://arxiv.org/pdf/1912.00049.pdf). In particular, they obtain the following average number of queries for untargeted attacks on ImageNet for eps=0.05 (taken from Table 2 from the [Square Attack paper](https://arxiv.org/pdf/1912.00049.pdf)):
| Method |  VGG | ResNet-50 |
| ------------- |-----------------------:| -----:|
| SignHunter | 95 | 129 |
| Square Attack | **31** | **73** |
| CorrAttack-Flip | 130 | 150 |
- Another related concern that I have is whether the proposed Bayesian optimization approach really helps. CorrAttack-Flip resembles a lot the SignHunter approach which doesn’t rely on taking the optimal action and instead selects actions in some predefined (but to some extent, arbitrary) order. Moreover, updates of SignHunter are similar in spirit: they also suggest to flip eps to -eps and vice versa, although they rely not on squares but on horizontal stripes (due to flattening of the image tensor). Thus, I would be very interested to see an ablation study for the success rate and query efficiency of (1) CorrAttack-Flip vs (2) CorrAttack-Flip with a **random selection of actions**. In other words, does the proposed Bayesian optimization framework really help, and if so, what is the margin? Related to this, there is an ablation study in Table 7 about the importance of PCA features, but the shown improvement is very minor and I am not sure if it’s statistically significant.


**Minor suggestions**
- It would be good to include at least a short discussion why you selected the expected improvement acquisition function and not other alternatives.
- Page 4: “where d is the image pixels” -> “where d is the **number** of image pixels”
- Implementation of the time-varying property: for completeness, it would be also good to provide an ablation study that would justify the choice out of the two alternatives you mentioned.
- All entries in Table 4 are based only on 10 samples. I’m not sure what kind of conclusion we can draw from an comparison which is made only based on 10 samples.

**Score:**
My current score is 4/10 but I would be willing to raise it if my second concern (mentioned in **Cons**) is resolved.

-----

**Update:**
After the rebuttal, I update my score to 6/10 (see the justifications below).

---

> ### Author Response · Authors · 2020-11-24
> **Response**
>
>
> Thanks for the comments. We address your main concerns as follows.
>
> 1. **Square Attack and SignHunter**. We have updated the comparison between CorrAttack, SignHunter and Square Attack in Table 1 and Table 2 (We remove the stripe initialization of Square Attack for fair comparison. Initialization is a disentangled part of the score-based attack. CorrAttack can also use the same initialization to reduce the queries).  While Square Attack and SignHunter are more query efficient on un-targeted attack of undefended model, CorrAttack needs lower number of queries on complicated task such as targeted attack and un-targeted attack of defended model. Un-targeted attack is relatively easy task and the flexible search space of Square Attack may have advantages. Search space is a complementary part of our method. CorrAttack may also be applied to the search space of Square Attack to model the correlation of each action to gain higher query efficiency for un-targeted on undefended model.
>
>    Targeted attack
>
>    | Success Rate    | VGG16      | Resnet50   | Densenet121 | Average Queries | VGG16    | Resnet50 | Densenet121 |
>    | --------------- | ---------- | ---------- | ----------- | --------------- | -------- | :------- | ----------- |
>    | SignHunter      | 93.52%     | 83.71%     | 90.75%      | SignHunter      | 2999     | 3905     | 3632        |
>    | Square Attack   | 96.69%     | 89.52%     | 95.38%      | Square Attack   | **2060** | 2807     | 2280        |
>    | CorrAttack-Flip | **98.07%** | **96.39%** | **99.41%**  | CorrAttack-Flip | 2191     | **2531** | **2019**    |
>
>    Un-targeted attack on the defense model
>
>    |                 | SignHunter | Square Attack | CorrAttack-Flip |
>    | --------------- | ---------- | ------------- | --------------- |
>    | Success Rate    | 68.97%     | 73.89%        | **79.15%**      |
>    | Average Queries | 1392       | 1086          | **1036**        |
>
> 2. **Ablation Study on CorrAttack-Flip Random.** We perform the ablation study on both CorrAttack-Flip Random and we have updated the result to the Appendix C.3. The result is also shown as follows, which shows that Bayesian optimization is effective for choosing the action.
>
>    Targeted Attack
>
>    | Success Rate           | VGG16      | Resnet50   | Densenet121 | Average Queries        | VGG16    | Resnet50 | Densenet121 |
>    | ---------------------- | ---------- | ---------- | ----------- | ---------------------- | -------- | -------- | ----------- |
>    | CorrAttack-Flip Random | 96.42%     | 92.92%     | 96.87%      | CorrAttack-Flip Random | 2545     | 3066     | 2556        |
>    | CorrAttack-Flip        | **98.07%** | **96.39%** | **99.41%**  | CorrAttack-Flip        | **2191** | **2531** | **2019**    |
>
>    Un-targeted Attack
>
>    | Success Rate           | VGG16    | Resnet50 | Densenet121 | Average Queries        | VGG16   | Resnet50 | Densenet121 |
>    | ---------------------- | -------- | -------- | ----------- | ---------------------- | ------- | -------- | ----------- |
>    | CorrAttack-Flip Random | **100%** | **100%** | **100%**    | CorrAttack-Flip Random | 143     | 176      | 132         |
>    | CorrAttack-Flip        | **100%** | **100%** | **100%**    | CorrAttack-Flip        | **130** | **150**  | **113**     |
>
>    Un-targeted Attack on Defense Model
>
>    |                 | CorrAttack-Flip Random | CorrAttack-Flip |
>    | --------------- | ---------------------- | --------------- |
>    | Success Rate    | 71.42%                 | **79.15%**      |
>    | Average Queries | 1159                   | **1036**        |
>
> 3. **PCA.** PCA is one feature that can be utilized to find the correlation of the reward and improve the query efficiency of the black-box attack. While the improvement is not significant, it is one useful feature of the GP. We believe other features may be added into the framework in the future (e.g. the embedding of the transfer-based attack like [TREMBA (ICLR 2020)](https://openreview.net/forum?id=SJxhNTNYwB)).
>
> 4. **Time varying alternative.** we perform the targeted attack on Resnet50. The success rate and average queries for removing samples near last action (line 2 of algorithm 2) are 95.70% and 2730 respectively. And the success rate and average queries of forgetting old samples (line 6 of algorithm 2) are 94.28% and 2845.
>
> 5. **Attacking Google Cloud Vision API.** The cost for attacking the Google Cloud Vision API is high and we cannot afford to attack many images for all the methods (about 10 US dollars for attacking 10 images). Experiment of 10 samples tries to show that CorrAttack is also efficient for real world classification model.

---

> > ### Comment · AnonReviewer2 · 2020-11-24
> > **Convincing rebuttal that resolves my main concerns**
> >
> > Thanks for the detailed answer.
> >
> > 1. **Square Attack and SignHunter**: the comparison to Square Attack and SignHunter is quite convincing in the sense that at least on some models the proposed method outperforms the existing methods. I agree that a specific initialization is important for boosting the query efficiency, although it's still unclear from the text what kind of initialization is used for the proposed method. In any case, one has to ensure that either we compare methods without *any* initialization whatsoever or with the best possible initialization for each method. The latter would be preferred, as I cannot fully agree that *"Initialization is a disentangled part"* -- there can still be some interplay between a method and its initialization. I would strongly suggest to include the results *with* the best possible initialization either in the rebuttal (if there is still time left for these experiments) or in the final version in case of acceptance.
> >
> > -----
> >
> > 2. **Ablation Study on CorrAttack-Flip Random.**: this ablation study also sounds convincing to me. But I would suggest to clearly describe its results in the main part: that choosing the action helps to improve the success rate and query efficiency, but not by a large amount.
> >
> > -----
> >
> > Overall, I am satisfied with the rebuttal and it resolves my both concerns. I update the score from 4/10 to 6/10. I think it's a clearly written paper and I hope that the additional results presented in the rebuttal will help to improve the paper and clearly frame its contributions (e.g., about the importance of the action selection).
> >
> > On the other hand, the paper presents an approach which is considerably more complicated than the current state-of-the-art methods like SignHunter or Square Attack, while achieving success rate and query efficiency which is better only on some models, but worse on others. That's why I am leaning towards 6/10 as the final score.

---

### Official Review · AnonReviewer3 · 2020-11-06
**Nice work on the path toward black-box adversarial attacks**

**Rating:** 6
**Confidence:** 3

**Review:**

This work considers an important problem of generating adversarial examples to attack a black-box model. The paper proposes a new approach to consider an adversarial example as a result of a sequence of pixel changes from a benign instance. Therefore, the adversarial generation problem can be considered as a bandit problem, and thus we can leverage Bayesian optimization to search for an instance that maximize the changes on the loss function through a sequence of pixel changes. The evaluation is comprehensive, and demonstrates that fewer number of black-box queries are needed to achieve a higher attack success rate.

I pretty much enjoy this work. The only concern is that I'm not so sure why the speed of CorrAttack is much faster than BayesOpt and Bayes-Attack. It seems that the main reason is because the work defines a block structure (as in Sec 4.1), so that the number of blocks is much smaller than the number of raw pixels. Such a hierarchical structure can naturally lead to a hierarchical search procedure as discussed in Sec 4.3. Is this the main reason why speed is fast? Such an idea doesn't seem to be unique to the bandit setup, is a similar idea also applicable to previous work such as BayesOpt and Bayes-Attack? I hope such an issue can be discussed more in the revision.

---

> ### Author Response · Authors · 2020-11-24
> **Response**
>
> Thanks for the comments. Please find our responses below.
>
> The main reason is that the GP of CorrAttack has much lower number of dimension. Consider a non-hierarchical version of CorrAttack. The time complexity of Gaussian process is O($dn^2$) where $n$ is the number of points or the number of queries. The running speed of these methods is largely determined by the $d$, which is the dimension of GP. BayesOpt and Bayes-Attack also shrink the dimension of image. However, the GP dimension of BayesOpt and Bayes-Attack is still much higher than CorrAttack. BayesOpt and Bayes-Attack search the overall perturbation directly. In contrast, CorrAttack defines a low dimension feature space, keep an overall perturbation and try to select an action on single block.
>
> For example, assume there are no hierarchical structure. CorrAttack searches over $24\times24\times3$ blocks and BayesOpt and Bayes-Attack also searches over the $24\times24\times3$ dimensional embedding.  The GP dimension $d$ of BayesOpt and Bayes-Attack is 1728 and the search space is continuous search space $[-eps,+eps]^{1728}$. However, the GP dimension of CorrAttack is only 4 with feature vector $z_{e_{ijk}}$ and the search space is discrete search space $E=\{e_{000}, e_{001}, \cdots, e_{hwc}\}$, whose number is 1728. CorrAttack has much lower dimension and it is the reason why CorrAttack is much faster.
>
> Hierarchical structure can accelerate CorrAttack at the initial stage but not the main reason that CorrAttack is faster. It is applicable to BayesOpt and Bayes-Attack but not very helpful since when dividing the blocks into smaller sizes, the GP dimension of BayesOpt and Bayes-Attack will increase from $d$ to $4d$. However, CorrAttack keeps $d=4$. Assume there are hierarchical structure and we divide the $24\times24\times3$ blocks into $48\times48\times3$ blocks. The GP dimension of BayesOpt and Bayes-Attack grow from 1728 to 6912 and the search space is continuous search space $[-eps,+eps]^{6912}$. However, the dimension of CorrAttack is still 4 with feature vector $z_{e_{ijk}}$ and the search space is discrete search space  $E=\{e_{000}, e_{001}, \cdots, e_{hwc}\}$, whose number is 6912.
>
> You may also look at the Appendix C.6 of the revision for more details.

---

### Decision · Program_Chairs · 2021-01-07
**Final Decision**

**Decision:**

Reject

**Comment:**

The paper presents a new Bayesian optimization method based on the Gaussian process bandits framework for black-box adversarial attacks. The method achieves good performance in the experiments, which was appreciated by all the reviewers.

At the same time, the presentation of the method is quite confusing, which currently precludes acceptance of the paper. In particular, during the discussion phase the reviewers were not able to decipher the algorithm based on the description presented in the paper. It is not clear how the problem is modeled as a bandit problem, what the loss function $\ell$ is minimized and why minimizing it makes sense (assuming, e.g., that $\ell$ it the hinge loss as suggested and the initial prediction is good with a large margin, that is, the loss is zero, equation 6 never changes $x_t$ when the procedure is started from $x$). This connection, since it is the fundamental contribution of the paper, should be much better explained. Once the problem is set up to estimate (maximize?) the reward, it is changed to calculating the difference in the minimization (cf. equation 11), which is again unmotivated. (Other standard aspects of the algorithm should also be explained properly, e.g., the stopping condition of Algorithm 1)

Unfortunately, the paper is written in a mathematically very imprecise manner. As an example, consider equation (6), where $B_p$ and the projection operator are not defined, and while these can be guessed, a projection of the argmin seems to be missing as well in the end (otherwise nothing guarantees that $x_T$, which is the final outcome of the algorithm, remains in the $L_p$ ball). Another example is the $Discrete\ Approximate\ CorrAttack_{Flip}$ paragraph which requires that every coordinate of $x$ should be changed by  $\pm\epsilon$. It is also not clear what "dividing the image into several blocks" means in Section 4.1 (e.g., are these overlapping, do they cover the whole image, etc., not to mention that previously $x$ was a general input, not necessarily an image). It is also unlikely that the stopping condition in Algorithm 1 would use the exact same $\epsilon$ for the acquisition function as the perturbation radius for adversarial examples, etc. While some of these inaccuracies and unclear definitions are also mentioned in the reviews, unfortunately there are more in the paper.

The authors are encouraged to resubmit the paper to the next venue after significantly improving and cleaning up the presentation.